# Chaperone-mediated ordered assembly of the SAGA and NuA4 transcription co-activator complexes in yeast

Alberto Elías-Villalobos [1], Damien Toullec[1], Céline Faux[1], Martial Séveno[2] & Dominique Helmlinger [1]*

Transcription initiation involves the coordinated activities of large multimeric complexes, but little is known about their biogenesis. Here we report several principles underlying the assembly and topological organization of the highly conserved SAGA and NuA4 co-activator complexes, which share the Tra1 subunit. We show that Tra1 contributes to the overall integrity of NuA4, whereas, within SAGA, it specifically controls the incorporation of the de-ubiquitination module (DUB), as part of an ordered assembly pathway. Biochemical and functional analyses reveal the mechanism by which Tra1 specifically interacts with either SAGA or NuA4. Finally, we demonstrate that Hsp90 and its cochaperone TTT promote Tra1 de novo incorporation into both complexes, indicating that Tra1, the sole pseudokinase of the PIKK family, shares a dedicated chaperone machinery with its cognate kinases. Overall, our work brings mechanistic insights into the assembly of transcriptional complexes and reveals the contribution of dedicated chaperones to this process.

[1] CRBM, CNRS, University of Montpellier, Montpellier, France. [2] BioCampus Montpellier, CNRS, INSERM, University of Montpellier, Montpellier, France. *email: dhelmlinger@crbm.cnrs.fr

A critical step in gene expression is transcription initiation, which is controlled by many factors that typically function as part of multimeric complexes. Genetic, biochemical and structural evidence indicates that their subunits form distinct modules with specific functions. Numerous studies have characterised their activities and regulatory roles in gene expression. In contrast, much less is known about how these complexes assemble, which chaperones are required, and whether their assembly can be modulated to control or diversify their functions. Deciphering these principles is important to fully understand their structural organisation, function and allosteric regulation[1]. Notably, chromatin-modifying and -remodelling complexes often share functional modules and therefore probably require dedicated mechanisms and chaperones for their proper assembly[2].

One such complex, the Spt-Ada-Gcn5 acetyltransferase (SAGA) co-activator, bridges promoter-bound activators to the general transcription machinery. In yeast, SAGA is composed of 19 subunits, which are organised into five modules with distinct regulatory roles during transcription[3,4]. These include histone H3 acetylation (HAT), histone H2B de-ubiquitination (DUB) and loading of TBP onto the promoter. A fourth module consists of a set of core components that scaffold the entire complex, most of which are shared with the general transcription factor TFIID. Finally, the largest SAGA subunit, Tra1, is shared with another transcriptional co-activator complex, yeast NuA4, which also contains a HAT module that preferentially targets histone H4 and the H2A.Z variant[5]. Tra1 directly interacts with a diverse range of transcription factors, and recruits SAGA and NuA4 to specific promoters upon activator binding (reviewed in refs. [6,7]).

Yeast Tra1 and its human ortholog, TRRAP, belong to a family of atypical kinases, the phosphoinositide 3 kinase-related kinases (PIKKs), but lack catalytic residues and are thus classified as pseudokinases[8–10]. The reason for the evolutionary conservation of a typical PIKK domain architecture within Tra1 orthologs is not known. Genetic and biochemical studies indicate that Tra1 primary role is to mediate the transactivation signal from activators by recruiting SAGA and NuA4 to chromatin. It has been difficult, however, to delineate the specific contribution of Tra1 to SAGA and NuA4 architecture and activities because, to date, no clear separation-of-function alleles exist. Indeed, the mechanism by which Tra1 interacts differentially with SAGA and NuA4 remains elusive.

The fission yeast Schizosaccharomyces pombe provides a unique opportunity to address this issue because it has two paralogous proteins, Tra1 and Tra2, and each has non-redundant roles that are specific for SAGA and NuA4, respectively[11]. Within SAGA, Tra1 has specific regulatory roles and does not contribute to its overall assembly[11], consistent with its peripheral position in the recent cryo-electron microscopy structure of SAGA from the budding yeast Pichia pastoris[12]. In contrast, a recent partial structure of the yeast NuA4 complex indicates that Tra1 occupies a more central position[13]. However, little is known about how Tra1 incorporates into the SAGA and NuA4 complexes, whether it involves similar or distinct mechanisms, and which chaperone or assembly factors are required.

Here, we show that, in S. pombe, Tra1 and Tra2 require Hsp90 and its cochaperone, the Triple-T complex (TTT), for their de novo incorporation into SAGA and NuA4, respectively. Furthermore, proteomic, biochemical and genetic approaches identify the residues mediating Tra1 specific interaction with SAGA. Notably, we demonstrate that Tra1 contacts a surprisingly small region of the core subunit Spt20, which is both necessary and sufficient for Tra1 interaction with SAGA. Kinetic analyses of nascent Tra1 incorporation reveal that it promotes the integration of the DUB module, uncovering an ordered pathway of SAGA assembly. Finally, in contrast to the specific role of Tra1 in SAGA architecture, we show that Tra2 has a general scaffolding role in NuA4 assembly. Overall, our work brings mechanistic insights into the assembly and modular organisation of two important transcriptional co-activator complexes.

## Results

**The TTT subunit Tti2 contributes to Tra1 and Tra2 functions.** Previous work in mammalian cells revealed that the Hsp90 cochaperone TTT stabilises PIKKs, including TRRAP, the human ortholog of yeast Tra1[14–19]. Three specific subunits, Tel2, Tti1 and Tti2, define the TTT complex in S. pombe, Saccharomyces cerevisiae and human cells (Supplementary Data 1)[16,20,21]. Some TTT subunits interact physically and genetically with Tra1 in S. cerevisiae and S. pombe[11,20–23]. Fission yeasts have two paralogous genes, tra1 and tra2, and each has non-redundant roles that are specific for SAGA or NuA4, respectively[11]. S. pombe thus offers a unique opportunity to study the specific contribution of TTT and Tra1 to SAGA and NuA4 organisation and function.

In human cells, TTI2 is critical for the stability of both TEL2 and TTI1 at steady state[17], in agreement with its stable binding to the TTT complex. We thus focused our investigations on Tti2, which we confirmed interacts with both Tra1 and Tra2 in S. pombe (Supplementary Data 1). Like Tti2, Tra2 is essential for viability, whereas tra1Δ mutants are viable[11,22]. We thus developed a strategy based on inducible CreER-loxP-mediated recombination to generate conditional knockout alleles of tti2 (tti2-CKO) and tra2 (tra2-CKO). These strains showed β-estradiol-induced loss of Tti2 or Tra2 expression, which correlated with progressive proliferation defects, but no obvious decrease in cell viability, at least within the time frame analysed (Supplementary Figs. 1, 2). Based on these observations, tti2-CKO and tra2-CKO strains were induced with β-estradiol for 18 and 21 h, respectively, before further analysis.

We then performed genome-wide expression analyses of DMSO- and β-estradiol-treated tti2-CKO and tra2-CKO cells, compared with a creER control strain treated identically. We also analysed tra1Δ mutants that were compared with a wild-type strain. Differential expression analysis revealed both specific and overlapping changes in each mutant (Fig. 1a–d; Supplementary Fig. 3). Specifically, gene expression changes correlated positively between tti2-CKO and tra2-CKO mutants, as well as between tti2-CKO and tra1Δ mutants (Fig. 1a). In contrast, no correlation was observed between tra2-CKO and tra1Δ mutants (Fig. 1a), as expected from their specific, non-redundant roles within either NuA4 or SAGA[11]. A Venn diagram of the most differentially expressed genes (FC ≥ 1.5, $P \leq 0.01$) shows the extent of the overlap between all three mutants (Fig. 1b). Remarkably, over half of the Tti2-dependent genes (105/184) are also regulated by Tra1 (25), Tra2 (72) or both Tra1 and Tra2 (8), suggesting that TTT has an important role in Tra1- and Tra2-dependent gene expression in S. pombe.

Hierarchical clustering of all differentially expressed genes ($P \leq 0.01$) identified at least seven distinct clusters (Supplementary Fig. 3b, c). These include a large cluster of transcripts in which levels decrease in both tti2-CKO and tra2-CKO mutants (Cluster 2, 527 genes), and a smaller one in which levels decrease in both tti2-CKO and tra1Δ mutants (Cluster 6, 172 genes). We independently measured the expression of genes from each cluster using quantitative RT-PCR analyses and confirmed that SPCC1884.01 and SPCAC977.12 levels (Cluster 2) decrease in both tti2-CKO and tra2-CKO mutants (Fig. 1d), while SCC569.05c and gst2 levels (Cluster 6) decrease in both tti2-CKO and tra1Δ mutants (Fig. 1c).

In S. cerevisiae, elegant biochemical and functional studies established that the primary role of Tra1 is to mediate activator-dependent recruitment of SAGA or NuA4 to specific promoters[24–30].

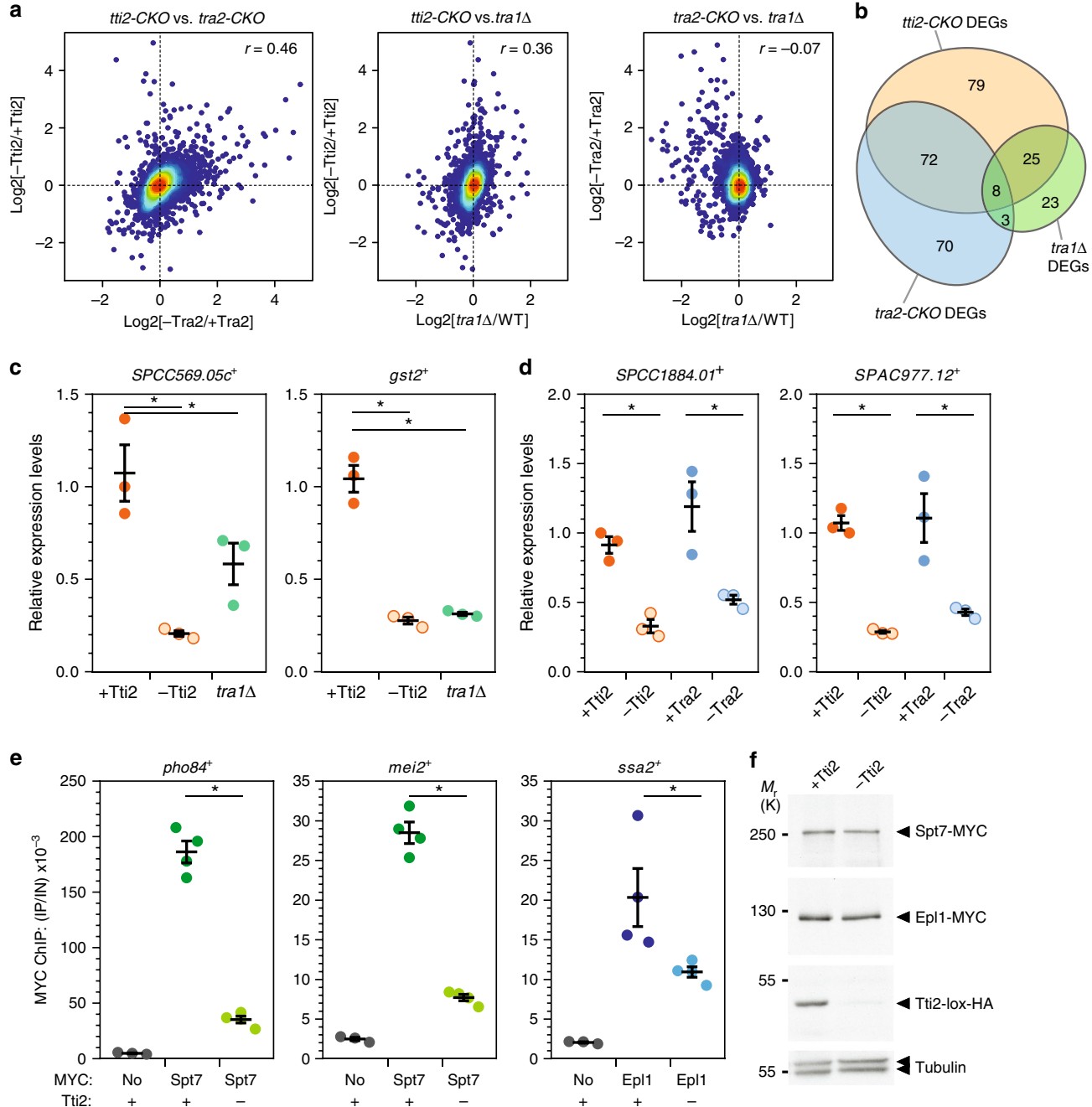

We thus evaluated the effect of Tti2 on the binding of the SAGA subunit Spt7 and the NuA4 subunit Epl1 to specific promoters, using chromatin immunoprecipitation (ChIP). Upon depletion of Tti2, we observed reduced occupancy of Spt7 at the *pho84* and *mei2* promoters and of Epl1 at the *ssa2* promoter, despite normal steady-state levels (Fig.1e, f).

In conclusion, we accumulated functional evidence that Tti2, likely as part of the TTT complex, contributes to the regulatory activities of Tra1 and Tra2 in gene expression. Therefore, similar to their active kinase counterparts, the Tra1 and Tra2 pseudokinases require the TTT cochaperone to function.

**Tti2 promotes Tra1 and Tra2 assembly into SAGA and NuA4.** These observations prompted us to test whether Tti2, as an Hsp90 cochaperone, promotes Tra1 and Tra2 incorporation into SAGA and NuA4, respectively, as shown for human mTOR and ATR-

containing complexes[16,18]. For this, we affinity purified SAGA and NuA4 upon *tti2* conditional deletion. Silver staining and quantitative MS analyses revealed a tenfold reduction of Tra1 from SAGA when Tti2 is depleted, as compared with control conditions (Fig. 2a). Similarly, we observed about a twofold reduction of Tra2 levels from NuA4 (Fig. 2b).

We next tested if Tti2 promotes the de novo incorporation of Tra1 into SAGA or, rather, prevents its disassembly. For this, we took advantage of the viability of *tra1Δ* mutants and disrupted the *tra1* promoter with a transcription terminator sequence flanked by *loxP* sites (*RI-tra1*, Supplementary Fig. 4a). With this allele, CreER-mediated recombination allows the inducible expression of Tra1 at endogenous levels. As a proof-of-principle, β-estradiol addition to *RI-tra1* strains suppresses their growth defects in conditions of replicative stress, using hydroxyurea (HU) (Supplementary Fig. 4b), to which *tra1Δ* mutants are sensitive[11]. Purification of SAGA from β-estradiol-treated

**Fig. 1** The TTT subunit Tti2 contributes to Tra1- and Tra2-dependent gene expression. **a**, **b** RNA-seq analyses of control *creER*, inducible *tti2* (*tti2-CKO*) and *tra2* knockouts (*tra2-CKO*), *tra1Δ* and wild-type (WT) strains (*n* = 3 independent biological samples). *creER*, *tti2-CKO* and *tra2-CKO* cultures were supplemented with either DMSO (+Tti2 or +Tra2) or β-estradiol (-Tti2 or -Tra2), for either 21 h (*creER* and *tra2-CKO*) or 18 h (*tti2-CKO*). **a** Density scatter plots comparing Tti2- with Tra2-depleted cells (left, *r* = 0.46, *P* < 0.001), Tti2-depleted cells with Tra1 deletion mutants (middle, *r* = 0.36, *P* < 0.001), and Tra2-depleted cells with Tra1 deletion mutants (right, *r* = −0.07, *P* < 0.001). Statistical significance and correlation were analysed by computing the Pearson correlation coefficient. Differential gene expression analyses were performed comparing cells treated with either DMSO (control) or β-estradiol (KO), while *tra1Δ* mutants were compared with isogenic WT cells. Genes whose expression is regulated by β-estradiol treatment (Supplementary Fig. 3a) were filtered out. **b** Venn diagrams showing the overlap of differentially expressed genes (DEGs) between all mutants. Using FC ≥ 1.5, *P* ≤ 0.01 thresholds, 184 DEGs were identified in Tti2-depleted cells, 153 in Tra2-depleted cells and 59 in *tra1Δ* mutants. Statistical significance was calculated using a hypergeometric test: *P* = 8.66e-115 comparing *tti2-CKO*, *tra2-CKO* and *tra1Δ*; *P* = 3.67e-72 comparing *tti2-CKO* with *tra2-CKO*; *P* = 2.37e-15 comparing *tti2-CKO* with *tra1Δ*; *P* = 0.004 comparing *tra2-CKO* with *tra1Δ*. **c**, **d** mRNA levels of the Tra1-dependent genes *SPCC569.05c* and *gst2* (**c**) and the Tra2-dependent genes *SPCC1884.01* and *SPAC977.12* (**d**) were measured using RT-qPCR of RNA extracted from *tti2-CKO*, *tra2-CKO* and *tra1Δ* strains grown as described in panels **a**, **b**. Each value represents mean mRNA levels from three independent experiments with the SEM. *act1* served as a control for normalisation across samples. Values from one control experiment were set to 1, allowing comparisons across culture conditions and strains. Statistical significance was determined by one-way ANOVA followed by Tukey's multiple comparison tests (asterisk: *P* < 0.05). **e** ChIP-qPCR was performed using *tti2-CKO* cells treated with either DMSO (+) or β-estradiol (−) for 18 h. ChIP of Spt7-MYC at the *pho84* and *mei2* promoters and of Epl1-MYC at the *ssa2* promoter serve as proxies for SAGA and NuA4 binding, respectively. A non-tagged strain was used as control for background IP signal (MYC: no). Ratios of MYC ChIP to input (IP/IN) from three independent experiments are shown as individual points, overlaid with the mean and SEM. Statistical significance was determined as in (**c**, **d**). **f** Anti-MYC and -HA western blotting of Tti2-HA, Spt7-MYC and Epl1-MYC in a fraction of the chromatin samples used for the ChIP-qPCR experiments shown in panel **e**. Equal loading was controlled using an anti-tubulin antibody. Source data are provided as a Source Data file

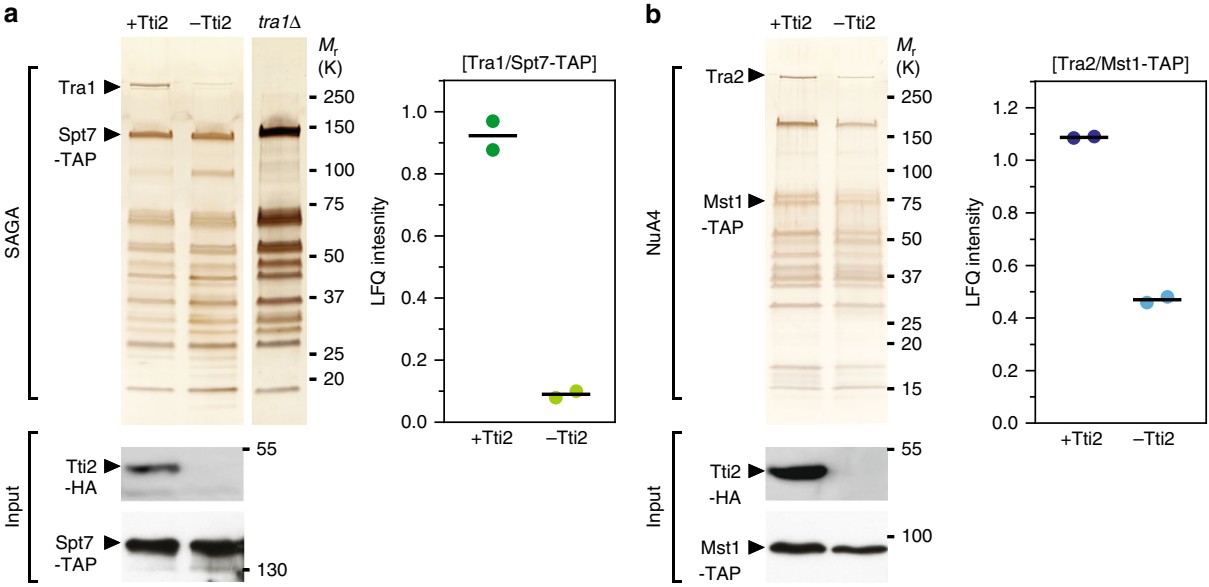

**Fig. 2** Tti2 promotes Tra1/Tra2 incorporation into SAGA/NuA4 complexes. **a** Silver staining of SAGA complexes purified either in the presence or absence of Tti2 (left). *spt7-TAP tti2-CKO* cells were grown to exponential phase in rich medium supplemented with either DMSO (+Tti2) or β-estradiol (−Tti2) for 18 h. SAGA was purified from a *tra1Δ* strain as a control for the complete loss of Tra1 from SAGA. LC-MS/MS analyses of SAGA purifications (right). LFQ intensity ratios of Tra1 to the bait, Spt7, from two independent experiments are plotted individually with the mean (black bar). Below are anti-HA western blotting of Spt7-TAP and Tti2-HA from a fraction of the input used for TAP. **b** Silver staining of NuA4 complexes purified either in the presence or absence of Tti2 (left). *mst1-TAP tti2-CKO* cells were grown to exponential phase in rich medium supplemented with either DMSO (+Tti2) or β-estradiol (−Tti2) for 18 h. LC-MS/MS analyses of NuA4 purifications (right). LFQ intensity ratios of Tra2 to the bait, Mst1, from two independent experiments are plotted individually with the mean (black bar). Below are anti-HA western blotting of Mst1-TAP and Tti2-HA from a fraction of the input used for TAP

*RI-tra1* cells showed a time-dependent, progressive increase of newly synthesised Tra1 (neo-Tra1) in Spt7 purification eluates, validating this approach for monitoring Tra1 de novo incorporation into SAGA (Fig. 3a). However, to conditionally deplete TTT from *RI-tra1* cells, we needed to develop a different strategy from the CreER-loxP-mediated knockout used so far. We attempted to tag Tti2 with an auxin-inducible degron (AID), but observed substantial destabilization even before adding the plant hormone auxin. In contrast, targeting another TTT complex subunit (Supplementary Data 1), Tel2, caused efficient and auxin-dependent degradation (Supplementary Fig. 5a–d). We then

analysed SAGA purifications from *RI-tra1 tel2-AID* cells using silver staining (Fig. 3b) and quantitative MS analyses (Fig. 3d). Both approaches showed decreased interaction between newly synthesised Tra1 and affinity purified Spt7 in cells partially depleted of Tel2. These results demonstrate that TTT contributes to the de novo incorporation of Tra1 into the SAGA complex.

Experiments with human cells suggested that TTT functions as an adaptor recruiting the HSP90 chaperone specifically to PIKKs[16,19,31]. We thus determined if Tra1 incorporation into SAGA requires Hsp90. We first tested the effect of conditionally inactivating Hsp90 on SAGA subunit composition at steady state.

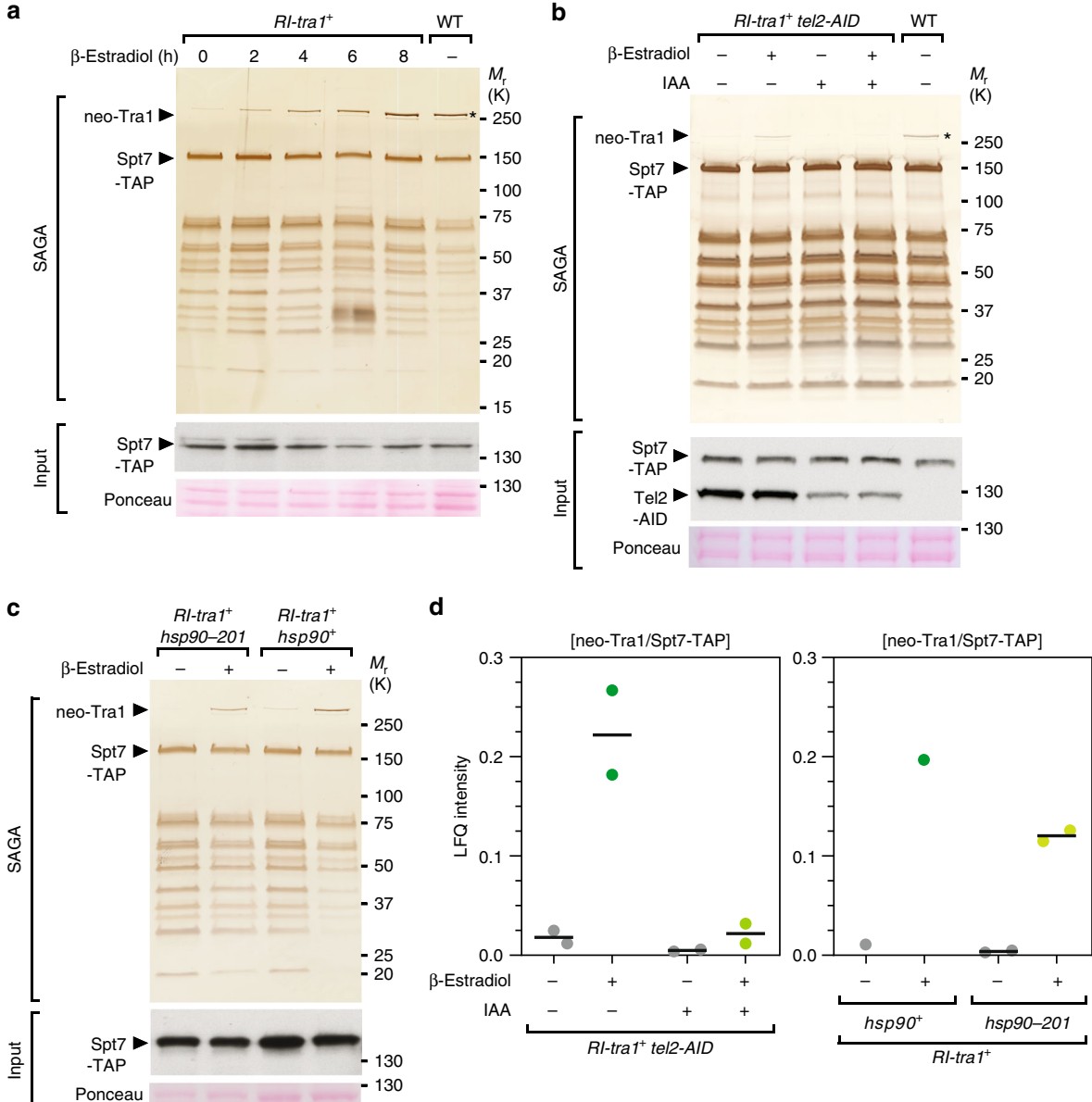

**Fig. 3** TTT and Hsp90 promote de novo assembly of Tra1 into SAGA. **a** Silver staining analysis of SAGA complexes purified upon Tra1 synthesis (neo-Tra1). *spt7-TAP RI-tra1* cells were grown to exponential phase and harvested at different time points after β-estradiol addition, as indicated (hours). SAGA was purified from a WT strain as a positive control, in which the asterisk labels steady-state Tra1 levels. Below is an anti-HA western blot of Spt7-TAP from a fraction of the input used for TAP. Ponceau red staining is used as loading control. Data are representative of four independent experiments. **b** Silver staining of SAGA complexes purified upon Tra1 synthesis (neo-Tra1), either in presence or absence of the TTT subunit Tel2. *spt7-TAP RI-tra1 tel2-AID* cells were grown to exponential phase in rich medium, supplemented with either ethanol (−IAA) or auxin (+IAA) for 16 h, and harvested 6 h after addition of either DMSO (−) or β-estradiol (+). SAGA was purified from a WT strain as a positive control, in which the asterisk labels steady-state Tra1 levels. Below are anti-HA western blotting of Spt7-TAP and Tel2-AID from a fraction of the input used for TAP. Both the TAP and AID sequences are in frame with HA epitopes. Ponceau red staining is used as loading control. Data are representative of two independent experiments. **c** Silver staining of SAGA complexes purified upon Tra1 synthesis (neo-Tra1), either in WT or *hsp90–201* mutant strains. *spt7-TAP RI-tra1 hsp90* and *spt7-TAP RI-tra1 hsp90–201* cells were grown to exponential phase in rich medium and harvested 6 h after addition of either DMSO (−) or β-estradiol (+). Below is an anti-HA Western blot of Spt7-TAP from a fraction of the input used for TAP. Ponceau red staining is used as loading control. Data are representative of two independent experiments. **d** LC-MS/MS analyses of SAGA purifications from control *spt7-TAP RI-tra1* strains, *spt7-TAP RI-tra1 tel2-AID* strains used in (**d**), and *spt7-TAP RI-tra1 hsp90–201* strains used in (**e**). LFQ intensity ratios of newly synthesised Tra1 (neo-Tra1) were normalised to the bait, Spt7. Ratios from two independent experiments are plotted individually with the mean (black bar). Source data are provided as a Source Data file

For this, we affinity purified Spt7 from *hsp90–26* temperature-sensitive mutants grown at either permissive or restrictive temperature[32]. Silver staining analysis showed that Hsp90 inactivation causes a specific decrease of Tra1 in Spt7 purification eluates (Supplementary Fig. 6). We next asked whether Hsp90 contributes to de novo incorporation of Tra1 into SAGA.

However, CreER cytoplasmic sequestration depends on a functional Hsp90. We thus used an *hsp90–201* strain, which harbors a weaker Hsp90 mutant allele[33]. Silver staining (Fig. 3c) and quantitative MS analyses (Fig. 3d) revealed a decrease of newly synthesised Tra1 in SAGA purified from *hsp90–201* mutants, as compared with wild-type cells. Although the observed

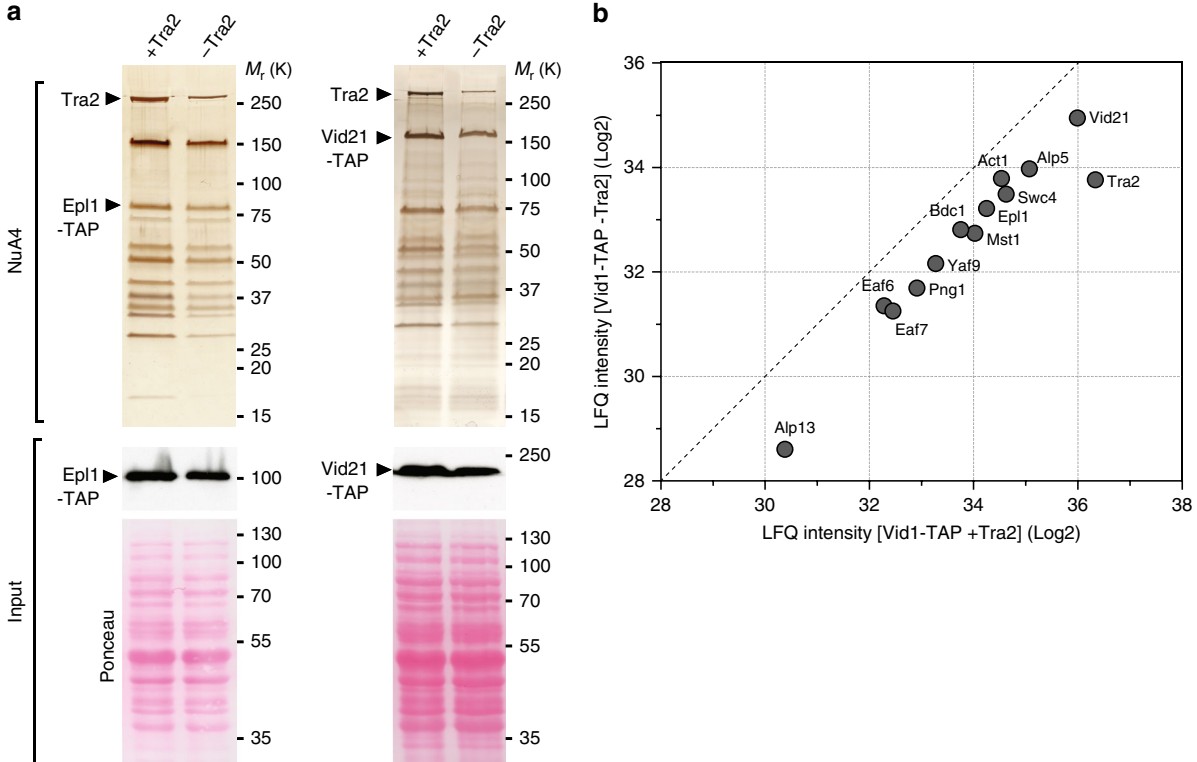

**Fig. 4** Tra2 is important for the formation of the entire NuA4 complex. **a** Silver staining of NuA4 complexes purified either in the presence or absence of Tra2, using Epl1-TAP (left) and Vid21-TAP (right) as baits. Cells were grown to exponential phase in rich medium supplemented with either DMSO (+Tra2) or β-estradiol (-Tra2), for either 16 h (*epl1-TAP tra2-CKO*) or 21 h (*vid21-TAP tra2-CKO*). Below are anti-HA western blotting of Epl1-TAP (left) and Vid21-TAP (right) from a fraction of the input used for TAP. Ponceau red staining is used as control for equal protein content and loading. **b** Scatter plot representing the LFQ intensities from LC-MS/MS analysis of NuA4 complexes purified in the presence (*x*-axis) or absence (*y*-axis) of Tra2. Individual points represent individual NuA4 subunits in Vid21-TAP eluates. The dashed line shows a 1:1 ratio. Data are representative of two independent experiments. Source data are provided as a Source Data file

effect is modest in this hypomorphic mutant, this observation supports the conclusion that Hsp90, like TTT, contributes to the de novo assembly of Tra1 into SAGA. Altogether, our results indicate that TTT acts as an Hsp90 cochaperone promoting the assembly of Tra1 into SAGA and likely Tra2 into NuA4. Therefore, pseudokinases and kinases of the PIKK family share a specific, dedicated chaperone machinery for their maturation and incorporation into active complexes.

**Tra1 and Tra2 have distinct roles within SAGA and NuA4.** We noted that the loss of TTT affected SAGA and NuA4 differently. Upon Tti2 depletion, the decrease of Tra1 does not affect the overall migration profile of SAGA, similar to what we observed in *tra1Δ* mutants (Fig. 2a)[11]. In contrast, the effect of Tti2 on Tra2 incorporation into NuA4 is less pronounced, but appears to cause a global decrease in the amount of purified NuA4 subunits (Fig. 2b), without affecting total protein content (Supplementary Fig. 1f). Alternatively, the bait used for this purification, the Mst1 HAT subunit, might dissociate from the rest of the complex upon *tti2* deletion and loss of Tra2 from NuA4. Finally, we noticed reduced Mst1 levels in total extracts from Tti2-depleted cells (Fig. 2b), possibly accounting for the decrease in the amount of purified NuA4.

To directly evaluate the effect of Tra2 on NuA4 subunit composition, we purified NuA4 upon *tra2* deletion, using *tra2-CKO* cells. For this, we affinity purified either Epl1, which anchors the HAT module to the rest of NuA4 in *S. cerevisiae*[34], or Vid21, whose *S. cerevisiae* ortholog Eaf1 forms a platform for

NuA4 assembly[35]. Silver staining and quantitative MS analyses of affinity purified Mst1, Epl1 and Vid21 revealed that each interact with a similar set of 13 proteins, which, altogether, define the NuA4 complex from *S. pombe* (Supplementary Fig. 7a–c), confirming and extending results from a previous study[21]. Then, upon Tra2 conditional depletion, we observed an overall decrease in the amount of purified NuA4 in both Epl1 and Vid21 purification eluates (Fig. 4a). Quantitative MS analyses of Vid21 purification eluates confirmed an overall decrease of all 13 NuA4 subunits when Tra2 is depleted (Fig. 4b). Importantly, cell viability and total protein content are not affected at this time point (Supplementary Fig. 2e, f; Fig. 4a). Furthermore, control SAGA purifications suggested that other chromatin-bound complexes can be efficiently extracted and purified upon Tra2 depletion (Supplementary Fig. 8). Altogether, these results indicate that, in contrast to Tra1 in SAGA, Tra2 contributes to the scaffolding and stabilisation of the entire NuA4 complex. Such distinct architectural roles provide a functional validation of the recent structural studies of yeast SAGA and NuA4, which showed that Tra1 occupies a peripheral position within SAGA and a more central position within NuA4[12,13,36–38].

**Mechanism of Tra1 specific interaction with SAGA.** We next sought to determine how Tra1 interacts specifically with SAGA, taking advantage of the viability of *tra1* mutants in *S. pombe* and guided by the most recent cryo-electron microscopy structure of SAGA from *P. pastoris* (Fig. 5a)[12]. Resolution of the secondary structure elements of Tra1 bound to SAGA identified a narrow

and highly flexible hinge region that was suggested to form the major, if not the single interaction surface between Tra1 and the rest of the complex[12]. This region is located near the start of the Tra1 FAT domain and consists of about 50 residues that fold into three distinct α-helices (H1-H3, Fig. 5a). Multiple alignments of Tra1 orthologs from yeast, invertebrate and vertebrate species indicate that this region is conserved throughout eukaryotes (Fig. 5a). Interestingly, the homologous region of *S. pombe* Tra2, which is only present in NuA4, is more divergent, suggesting that this region might dictate SAGA binding specificity.

Deletion of a few helices within Tra1 can cause important structural rearrangements and destabilise the protein[30]. Thus, to determine the contribution of this 50-residue region to Tra1-SAGA interaction, we swapped them with those from the closest homolog of Tra1, *S. pombe* Tra2, which does not interact with SAGA[11]. We also introduced the corresponding sequence from *S. cerevisiae* Tra1 (Fig. 5b), which is shared between SAGA and NuA4. We first verified that both Tra1-SpTra2 and Tra1-ScTra1 mutant proteins are expressed at levels similar to those of wild-type Tra1 (Fig. 5c). In contrast, silver staining and quantitative MS analyses revealed that the Tra1-SpTra2 hybrid is not detectable in Spt7 purifications, whereas normal levels of the Tra1-ScTra1 hybrid are observed (Fig. 5c). Similarly, a Tra1-mTOR hybrid protein is unable to copurify with SAGA (Supplementary Fig. 9a), consistent with human mTOR assembling into unrelated complexes, TORC1 and TORC2. Importantly, quantitative MS analyses show that both Tra1-SpTra2 and Tra1-ScTra1 hybrid mutant proteins efficiently copurify with Tti2 (Supplementary Fig. 9b). Thus, this region does not affect Tra1 binding to TTT and the Tra1-SpTra2 mutant protein is recognised by its cochaperone, despite being unable to interact with SAGA.

Phenotypic analyses of *tra1-Sptra2* and *tra1-Sctra1* strains showed that *tra1-Sptra2* mutants are sensitive to HU and caffeine, similar to *tra1Δ* mutants, whereas *tra1-Sctra1* strains show no growth defects, as compared with wild-type cells (Fig. 5d). RNA-seq analyses of *tra1-Sctra1* and *tra1-Sptra2* mutants revealed that the transcriptomic changes observed in *tra1-Sptra2* and *tra1Δ* mutants correlate well ($r^2 = 0.58$), as compared with wild-type cells (Fig. 5e). In contrast, *tra1-Sctra1* mutants show little changes, correlating poorly with *tra1Δ* mutants ($r^2 = 0.16$) (Fig. 5f).

To conclude, a 50-residue region from *S. cerevisiae* Tra1 complements the orthologous region from *S. pombe* Tra1, likely because *S. cerevisiae* Tra1 is present in both SAGA and NuA4. In contrast, the paralogous region from *S. pombe* Tra2 diverged such that it cannot interact with SAGA. Whether this region of Tra2 is responsible for its specific integration into NuA4 remains to be determined. Altogether, structural, biochemical and functional evidence demonstrates that Tra1 directly contacts SAGA through a restricted, 50-residue region located at the beginning the FAT domain. This region of Tra1 consists of three α-helices that fold into a cup-shaped structure (Fig. 5a)[12], which we thus coined the Cup SAGA Interacting (CSI) region of Tra1.

**The SAGA subunit Spt20 anchors Tra1 into the SAGA complex.** Patrick Schultz's laboratory reported that the Tra1-SAGA hinge accommodates a putative α-helix belonging to a SAGA subunit other than Tra1[12]. This observation encouraged us to identify the residues forming the other side of the hinge and directly contacting the Tra1 CSI region. Besides Tra1, 18 subunits form the *S. pombe* SAGA complex[39]. Genetic, biochemical and structural evidence suggests that of these, Ada1, Taf12 and Spt20 are candidates to anchor Tra1 within SAGA[11,12,36,37,40,41]. Silver staining analyses revealed that Tra1 is undetectable in SAGA

purified from *spt20Δ* mutants, without any other visible changes in its overall migration profile (Fig. 6a). Spt20 is therefore essential for Tra1 incorporation into SAGA.

*S. pombe* Spt20 is 474-residue long and comprises an N-terminal half that contains several conserved regions, named homology boxes (HB)[42], and a C-terminal low-complexity region (LCR) (Fig. 6b). Deletion of the Spt20 N-terminal half (residues 1–255) abolished its interaction with SAGA (Supplementary Table 1), indicating that this portion of Spt20 mediates its integration into the complex. Silver staining analyses of SAGA purified from mutants that remove various lengths of the Spt20 C-terminal LCR identified a short region of 11 residues that is crucial for Tra1-SAGA interaction (Fig. 6c). Quantitative MS analyses confirmed that Tra1 does not interact with SAGA in *spt20–290* mutants, in which residues 291–474 are deleted, whereas normal levels of Tra1 are detected in *spt20–300* mutants, in which residues 301–474 are deleted (Fig. 6c).

Structure prediction identified an α-helix in this region, which we coined the head interacting with Tra1 (HIT) (Fig. 6d). Silver staining and quantitative MS analyses of SAGA purified from mutants in which the Spt20 HIT region is deleted (*spt20-HITΔ*) confirmed its importance for Tra1 interaction (Fig. 6e). Similarly, mutating the HIT identified four residues, FIEN, that are important for Tra1 incorporation into SAGA, whereas the four positively charged RRKR residues contribute less (Fig. 6e). We verified that all Spt20 truncation, deletion and point mutants are expressed at levels comparable to those of wild-type Spt20 (Supplementary Fig. 10a) and, importantly, are present in purified SAGA complexes (asterisk in Fig. 6c, e). Furthermore, in all these mutants, Tra1 is expressed at levels similar to those observed in WT cells (Supplementary Fig. 10b). This observation suggests that unassembled Tra1 is stable, contrary to core SAGA subunits[40], and folded correctly, in agreement with the similarity of Tra1 structures whether alone or within SAGA[7].

We next evaluated the phenotype of *spt20-HIT* mutant strains. Similar to *tra1Δ* mutants, *spt20-HITΔ* and *spt20-FIEN* mutants are sensitive to HU, whereas *spt20-RRKR* show milder defects, as compared with wild-type cells (Fig. 6f). RNA-seq analyses of *spt20Δ* and *spt20-HITΔ* mutants revealed similar transcriptomic changes to those observed in *tra1-Sptra2* and *tra1Δ* mutants, as compared with a wild-type strain. Comparing *spt20-HITΔ* with *spt20Δ* mutants revealed that the Spt20 HIT region contributes to the expression of only a subset of Spt20-dependent genes ($r^2 = 0.35$) (Fig. 6g), consistent with the HIT region being specifically involved in Tra1 interaction. Indeed, we observed a better correlation between *spt20-HITΔ* and *tra1Δ* mutants ($r^2 = 0.44$) (Fig. 6h). Remarkably, the strongest correlation was obtained when comparing *spt20-HITΔ* with *tra1-Sptra2* mutants ($r^2 = 0.62$) (Fig. 6i), i.e., strains in which the hinge is mutated on either side of the same interaction surface. Altogether, these biochemical and functional approaches identified a narrow region of Spt20 that is necessary to integrate Tra1 into SAGA, likely by direct interaction with the Tra1 CSI region.

**Spt20 is necessary for Tra1 incorporation in *S. cerevisiae*.** Such a restricted and specific interaction surface might have appeared specifically in *S. pombe*, because Tra1 and Tra2 diverged enough to interact exclusively with either SAGA or NuA4. We thus sought to identify a homologous Spt20 HIT region in *S. cerevisiae*, in which a single Tra1 protein can interact with both SAGA and NuA4.

Although *S. pombe* and *S. cerevisiae* Spt20 orthologs diverged substantially[42], their overall domain organisation is similar. We thus focused our mutational analysis of *S. cerevisiae* Spt20 to a region located between the HB and the LCR (Fig. 7a). Silver

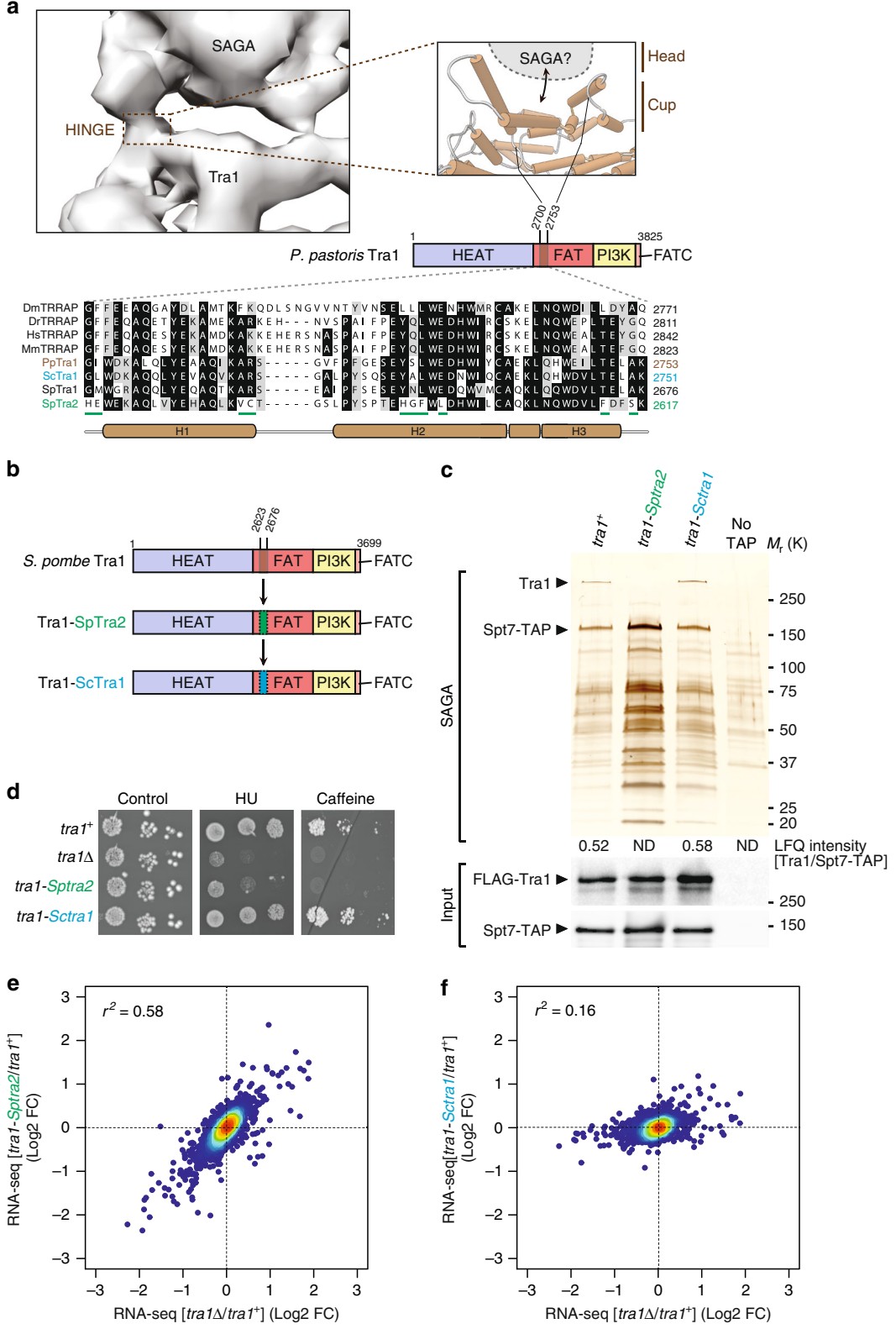

staining analyses of *S. cerevisiae* SAGA purified from mutants that remove various lengths of the Spt20 C-terminal LCR identified a short region of 18 residues (474–492) that is critical to incorporate Tra1 into SAGA (Fig. 7b). Quantitative MS analyses confirmed the importance of this region for Tra1 interaction with SAGA. Indeed, <3% of Tra1 is detected in Spt7 purification eluates from *spt20–474* mutants, as compared with *spt20–492* mutants or WT controls (Fig. 7c). Western blotting of Spt20 truncation mutants confirmed that each mutant is still present in SAGA (Fig. 7d). Finally, secondary structure prediction identified a α-helix in this region (Fig. 7a), further arguing that the Spt20 HIT region is functionally and structurally conserved between *S. pombe* and *S. cerevisiae*.

**Fig. 5** Mechanism of Tra1 specific interaction with SAGA. **a** Close-up view of the putative region of Tra1 that contacts the rest of SAGA, which constitutes the flexible hinge in the structure of *P. pastoris* SAGA (EMD: 3804). Cartoon cylinders represent α-helices in *P. Pastoris* Tra1 structure (PDB: 5OEJ). This domain is located near the start of the FAT domain and corresponds to residues 2700–2753 (brown-coloured box). A homologous region, defined as the Cup SAGA Interacting (CSI), was identified in *S. pombe* Tra1 (residues 2623–2676) from multiple alignments of Tra1 orthologs, shown at the bottom. Residues that appear unique to *S. pombe* Tra2 are underlined (green). **b** Schematic illustration of the hybrid mutant alleles of *S. pombe* tra1+ that were constructed. Residues 2623–2676 from *S. pombe* Tra1 were swapped with the homologous region from either *S. pombe* Tra2 (green, residues 2564–2617), to create the *tra1-Sptra2* allele, or *S. cerevisiae* Tra1 (blue, residues 2698–2751), to create the *tra1-Sctra1* allele. **c** Silver staining of SAGA complexes purified from WT, *tra1-Sptra2* and *tra1-Sctra1* strains (see **b**), using Spt7 as the bait. A non-tagged strain (no TAP) was used as a control for background. Numbers at the bottom of the gel represent LFQ intensity ratios of Tra1 to the bait, Spt7, from LC-MS/MS analyses of purified SAGA complexes (ND: not detected or <1% of WT). Below are anti-FLAG and anti-HA western blotting of FLAG-Tra1 and Spt7-TAP in a fraction of the input used for TAP. Data are representative of five independent experiments. **d**–**f** HU sensitivity (**d**) and gene expression changes (**e**, **f**) of *tra1-Sptra2* and *tra1-Sctra1* strains, as compared with *tra1Δ* mutants. **d** Tenfold serial dilutions of exponentially growing cells of the indicated genotypes were spotted on rich medium (control), medium supplemented with 10 mM HU, or 15 mM caffeine, and incubated at 32 °C. **e**, **f** Density scatter plots from RNA-seq data comparing *tra1Δ* mutants (*x*-axis) with either *tra1-Sptra2* mutants (*y*-axis in **e**) or *tra1-Sctra1* mutants (*y*-axis in **f**), relative to isogenic WT controls (*n* = 3 independent biological samples). Statistical significance and correlation were analysed by computing the Pearson coefficient of determination ($r^2$ = 0.58 for *tra1-Sptra2* vs. *tra1Δ* and $r^2$ = 0.16 for *tra1-Sctra1* vs. *tra1Δ*; *P* < 0.0001). Source data are provided as a Source Data file

**The Spt20 HIT region is sufficient for Tra1 interaction**. We next asked whether the Spt20 HIT region is sufficient to interact with Tra1. For this, a peptide of about 50 residues encompassing the HIT region from either *S. pombe* or *S. cerevisiae* was immobilised on a column, through fusion to GST, and incubated with *S. pombe* protein extracts prepared from wild-type, *tra1-Sptra2* and *tra1-Sctra1* strains. We observed that both recombinant Spt20 HIT fragments specifically pull-down wild-type *S. pombe* Tra1, as compared with GST alone (lanes 1 vs. 2, Fig. 7e). Similar levels of the Tra1-ScTra1 hybrid are recovered on the GST-HIT column (lanes 5 vs. 6, Fig. 7e). Consistent with our in vivo observations that the Tra1 CSI region mediates this interaction, lower amounts of the Tra1-SpTra2 hybrid mutant are recovered on the GST-HIT column (lane 3 vs. 4, Fig. 7e). Overall, these experiments indicate that the Spt20 HIT region folds into a α-helix that is both necessary and sufficient for anchoring Tra1 in two highly divergent yeast species. We have thus elucidated the structural elements forming the narrow hinge and mediating the specific contact between Tra1 and the rest of SAGA. Notably, only a few residues are involved, in agreement with the peripheral position of Tra1 within SAGA[12].

**Tra1 orchestrates an ordered pathway for SAGA assembly**. Throughout this study, quantitative MS analyses of *S. pombe* SAGA purified from various mutants revealed an unexpected finding: the amount of DUB module subunits within SAGA consistently decreased when Tra1 is unassembled. For instance, we measured a reproducible decrease of both Sgf73 and Ubp8 in Spt7 purified from *tra1Δ* or β-estradiol-treated *tti2-CKO* cells, as compared with isogenic controls (Fig. 8a, b). Similarly, mutating either side of the hinge reduces the amount of both Sgf73 and Ubp8 in SAGA purifications, as observed in *spt20–290*, *spt20-HITΔ*, *spt20-FIEN* and *tra1-Sptra2* mutants (Hinge in Fig. 8a, b) (Supplementary Table 2). In contrast, the levels of Sgf73 and Ubp8 do not change in Spt7 purifications from *spt20–300*, *spt20-RRKR* and *tra1-Sctra1* strains, in which Tra1 incorporates into SAGA (Supplementary Table 2). The other two DUB subunits, Sgf11 and Sus1, are ~10 kDa, and therefore less reliably quantified by MS. The reproducibility of this effect across distinct mutants that all affect Tra1 incorporation suggested that Tra1 promotes assembly of the DUB module into SAGA.

We first asked whether, conversely, the DUB module stabilises Tra1 within SAGA. In *S. cerevisiae*, Sgf73 is critical to anchor the DUB module into SAGA[43]. Mass spectrometry analyses confirmed that, in *S. pombe sgf73Δ* mutants, the DUB subunits Ubp8, Sgf11 and Sus1 are absent from SAGA purifications

(Supplementary Table 3). In contrast, silver staining and MS analyses of SAGA indicated that Spt20 and Tra1 assembly were unaffected in *sgf73Δ* mutants (Fig. 8c). Altogether, these results suggest that SAGA assembly follows a unidirectional, ordered pathway, in which Spt20 anchors Tra1, which then stabilises the DUB module.

To test this hypothesis directly, we purified SAGA from *RI-tra1* strains (Fig. 3a), in which endogenous Sgf11 is tagged with a MYC epitope. We then concomitantly monitored the assembly kinetics of both newly synthesised Tra1 and the DUB module. We first confirmed that Sgf11 interacts less strongly with Spt7 in *tra1Δ* mutants or untreated *RI-tra1* cells, as compared with control conditions (lanes 2 and 3 vs. 6, Fig. 8d). We then observed a progressive increase in the amount of Sgf11 interacting with Spt7 upon β-estradiol treatment and de novo Tra1 incorporation (Fig. 8d). Quantification of four independent experiments confirmed the reproducibility of these observations (Fig. 8e). Supporting the existence of an ordered assembly pathway, comparing the relative levels of Sgf11 and Tra1 in SAGA at 4 h of induction shows that integration of the DUB module is slightly delayed compared with Tra1.

Overall, we accumulated functional and biochemical evidence supporting a model in which nascent Tra1 is recognised by the Hsp90 cochaperone TTT, possibly to catalyse its folding into a mature conformation. Tra1 is assembled by direct interaction with a small region of Spt20 and then promotes the incorporation of the DUB module within SAGA (Fig. 8f).

## Discussion

Many chromatin and transcription regulators function within large multimeric complexes. Deciphering the principles that govern their assembly is key to understanding their structural organisation, function and regulation. Our work brings several mechanistic insights into the de novo assembly and modular organisation of two such complexes, SAGA and NuA4. First, the Hsp90 cochaperone TTT promotes Tra1 and Tra2 incorporation into SAGA and NuA4, respectively. Second, structure-guided mutational analyses elucidated the specificity of Tra1 interaction with SAGA *vs.* NuA4. The topology of the Tra1-SAGA interaction surface consists of a small region of the Tra1 FAT domain contacting a single α-helix of the core subunit Spt20. Third, in contrast to the general role of Tra2 in NuA4 complex formation, Tra1 specifically controls the incorporation of the DUB module into SAGA, uncovering an ordered pathway of SAGA assembly.

We show here that SAGA and NuA4 require a dedicated chaperone machinery to be fully assembled. A recent study in

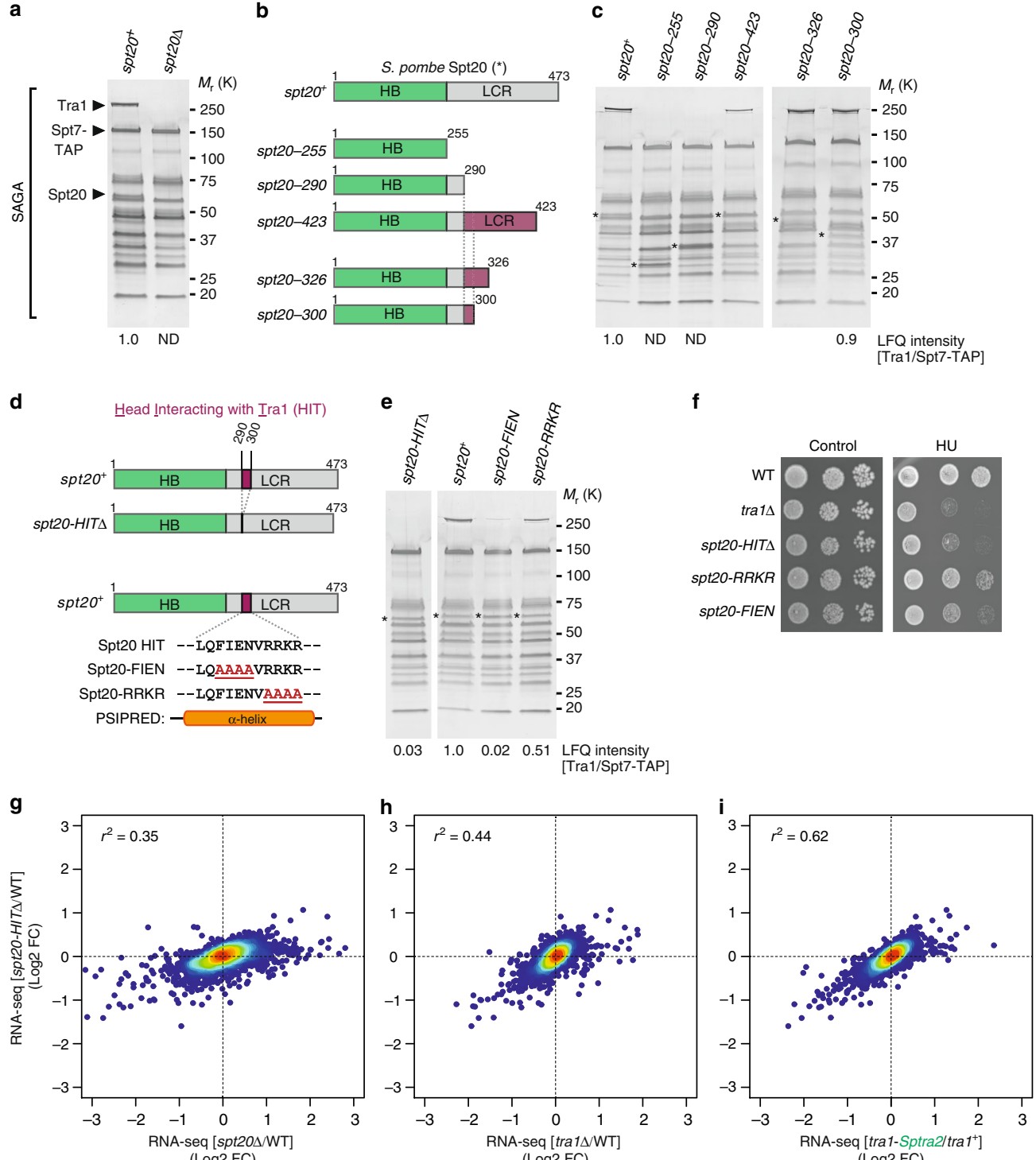

human cells showed that both the TFIID-specific subunit TAF5 and its SAGA-specific paralog TAF5L require a specific chaperone, the CCT chaperonin, for their incorporation into pre-assembled modules[44]. Our work therefore contributes to the emerging concept that dedicated chaperone machineries and ordered pathways control the de novo assembly of chromatin- and transcription-regulatory complexes.

Studies in mammals revealed that the pleiotropic HSP90 chaperone is specifically recruited to PIKKs by a dedicated cochaperone, the TTT complex, to promote their stabilisation and assembly into active complexes[14–19]. In contrast, the effect of TTT on the Tra1 pseudokinase, the only inactive member of the PIKK family, is less characterised. TTT stabilises TRRAP in human cells[14,17–19] and several studies reported physical and genetic interaction between Tra1 and TTT components in yeast[11,20–23]. We accumulated functional and biochemical evidence that, in S. pombe, Hsp90 and TTT promote the incorporation of Tra1 and Tra2 into the SAGA and NuA4 complexes, respectively. In agreement, we found that the TTT subunit Tti2 contributes to Tra1- and Tra2-dependent gene expression, as well

**Fig. 6** Spt20 anchors Tra1 into the SAGA complex. **a** Silver staining of SAGA complexes purified from WT and spt20Δ strains, using Spt7 as the bait. A band corresponding to Spt20 disappears in spt20Δ mutants. **b** Schematic illustration of the different Spt20 truncation mutants constructed to identify the Head Interacting with Tra1 (HIT) region of Spt20 from *S. pombe*. Distinct colours depict Spt20 domains, defined as Homology Boxes (HB) and a Low Complexity Region (LCR). Each mutant is named after to the last residue present in the truncation mutant, which shortens the LCR to various extent, as illustrated. **c** Silver staining of SAGA complexes purified from WT and spt20 truncation mutants, using Spt7 as the bait. Numbers at the bottom of the gel represent LFQ intensity ratios of Tra1 to Spt7, from LC-MS/MS analyses of purified SAGA complexes (ND: not detected or <1% of WT). Purple colouring depicts the region of Spt20 required for Tra1-SAGA interaction as inferred from each truncation mutant. Asterisks indicate the position of the WT and mutant Spt20 proteins in purified SAGA. **d** Schematic illustration of the deletion or point mutations within Spt20 HIT region, narrowed down to residues 290–300 (purple-coloured box) and predicted to fold into a α-helix, using PSI-blast based secondary structure PREDiction (PSIPRED[70]). **e** Silver staining of SAGA complexes purified from WT, spt20-HITΔ, spt20-FIEN and spt20-RRKR mutants, using Spt7 as the bait. Numbers at the bottom of the gel represent LFQ intensity ratios of Tra1 to Spt7, from LC-MS/MS analyses of purified SAGA complexes. Values for each mutant are expressed as percentage of WT SAGA. Asterisks indicate the position of the WT and mutant Spt20 proteins in purified SAGA. **f** HU sensitivity of spt20-HITΔ, spt20-FIEN and spt20-RRKR mutants, as compared with tra1Δ mutant strains. Tenfold serial dilutions of exponentially growing cells of the indicated genotypes were spotted either on rich medium (control) or medium supplemented with 5 mM HU and incubated for 3 days at 32 °C. **g–i** Density scatter plots plots from RNA-seq data comparing spt20-HITΔ mutants (y-axis) with either spt20Δ mutants (x-axis in **g**), tra1Δ mutants (x-axis in **h**), or tra1-Sptra2 mutants (x-axis in **i**), relative to isogenic WT controls (n = 3 independent biological samples). Statistical significance and correlation were analysed by computing the Pearson coefficient of determination ($r^2 = 0.35$ for spt20-HITΔ vs. spt20Δ; $r^2 = 0.44$ for spt20-HITΔ vs. tra1Δ; $r^2 = 0.62$ for spt20-HITΔ vs. tra1-Sptra2; $P < 0.0001$). Source data are provided as a Source Data file

as SAGA and NuA4 promoter recruitment. Therefore, although Tra1 is the sole catalytically inactive member of the PIKK family, it shares a dedicated chaperone machinery with active PIKK kinases for its folding, maturation, and assembly into a larger complex.

Phylogenetic analyses of PIKK orthologs in various organisms indicate that the Tra1 pseudokinase appeared early in the eukaryotic lineage, concomitantly with other PIKKs (our unpublished observations). As expected for a pseudokinase, catalytic residues diverged substantially. However, Tra1 orthologs retain the distinctive domain architecture of all PIKKs, which consists of a long stretch of helical HEAT repeats, followed by TPR repeats forming the FAT domain, preceding the FRB, PI3K-like and FATC domains. It is thus tempting to speculate that the requirement of PIKKs for a dedicated cochaperone explains the selection pressure that is observed on the sequence and domain organisation of Tra1, in the absence of conserved, functional catalytic residues. For example, the short, highly conserved C-terminal FATC domain loops back close to the active site and is critical for mTOR kinase activity[45]. Similarly, we found that the FATC domain is essential for Tra1 incorporation into SAGA (our unpublished observations), perhaps through allosteric control of the folding and positioning of the CSI region, which directly contacts SAGA.

Biochemical and functional evidence suggested that the Tra1 pseudokinase serves as a scaffold for the assembly and recruitment of the SAGA and NuA4 complexes to chromatin. *S. pombe* provides a unique opportunity to better understand its roles within each complex because it has two paralogous proteins, Tra1 and Tra2, and each has non-redundant functions that are specific for SAGA and NuA4, respectively[11]. Our work indicates that, within SAGA, Tra1 has specific regulatory roles and does not scaffold the entire complex but, rather, controls the assembly of the DUB module. In contrast, Tra2 contributes to the overall integrity of NuA4. In agreement, the most recent structures of yeast SAGA and NuA4 showed different positions of Tra1 relative to other subunits. Within SAGA, Tra1 localises to the periphery of the complex[12] and directly interacts with Spt20 (Figs. 6, 7), whereas it occupies a more central position within NuA4 and contacts several different subunits[13]. We therefore anticipate that the single Tra1 protein found in most other eukaryotic organisms has distinct architectural roles between SAGA and NuA4 and functions as a scaffold only within the NuA4 complex.

However, what determines the distribution of Tra1 between SAGA and NuA4 remains elusive. One possibility is that SAGA-

and NuA4-specific subunits compete for binding to the Tra1 CSI region. This mechanism would be similar to that described for mTOR assembly into the TORC1 and TORC2 complexes. Their structures revealed that the TORC1-specific subunit Raptor and the TORC2-specific subunit Rictor compete for binding to the same HEAT repeats of mTOR[46,47]. Similarly, electron microscopy and cross-linking coupled to MS indicate that the Tra1 FAT domain makes extensive contacts with several distinct NuA4 subunits[13,38]. We speculate that these interactions would sterically hinder the binding of Spt20 to the three α-helices forming the Tra1 CSI region (Fig. 5a). Higher resolution structures and further biochemical studies are required to test this hypothesis and explain why Tra1 binding to SAGA and NuA4 is mutually exclusive.

In marked contrast with *S. cerevisiae* and mammals, a tra1Δ deletion mutant is viable in *S. pombe*, enabling detailed biochemical and genetic studies that are more difficult in other organisms[6]. Taking advantage of this opportunity, we made significant progress in our understanding of the topological organisation of the Tra1-SAGA interface. The latest structure of SAGA clearly shows that Tra1 occupies a peripheral position and interacts with the rest of the complex through a narrow and flexible surface interaction, forming a hinge[12]. Our structure–function analyses identified the residues that constitute the hinge. Specifically, a small 50-residue region of the large Tra1 protein dictates the specificity of its interaction with SAGA. The homologous region from *S. pombe* Tra2 diverged such that it cannot interact with SAGA. Conversely, within the hinge, a density predicted to form a α-helix not attributable to Tra1 was observed at the threshold used to resolve Tra1 secondary structure elements[12]. Here, we demonstrate that a short portion of Spt20, which we named the HIT region, is both necessary and sufficient to anchor Tra1 within SAGA. The strong dependency of Tra1 on Spt20 HIT region (Figs. 6, 7) suggests that these residues constitute the main interface between Tra1 and the rest of SAGA, allowing the construction of unique separation-of-function mutations for phenotypic and functional analyses. The exact roles of Tra1/TRRAP have been challenging to study genetically because of its presence in both SAGA and NuA4[30] and of its essential roles in *S. cerevisiae* proliferation or during mouse early embryonic development[9,48]. As shown in *S. cerevisiae* (Fig. 7), identifying the residues mediating Tra1-SAGA interaction paves the way for addressing this issue.

Previous work suggested that Ada1 contributes to Tra1 incorporation into SAGA[40] and that Taf12 might form part of the

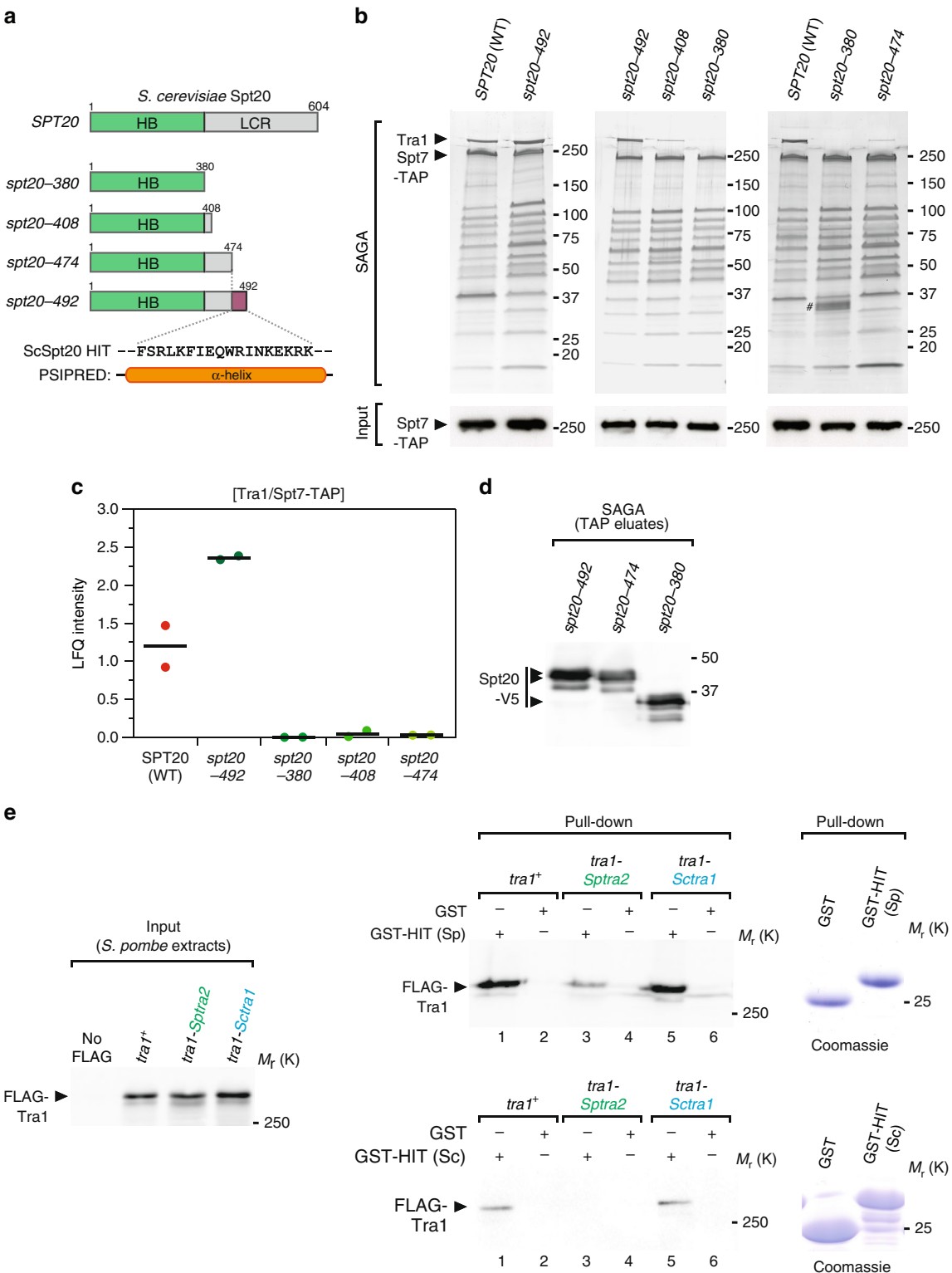

hinge[12,36]. Biochemical and functional evidence indicates that these subunits heterodimerize, as part of an octamer of histone folds forming the structural core of SAGA, analogous to that of TFIID[2]. Even if specific residues of Ada1 and/or Taf12 directly contact Tra1 and stabilise its interaction with SAGA, we predict that their contribution will be minor, at least under the experimental conditions tested here. Our results demonstrate that the Spt20 HIT region is both necessary and sufficient for Tra1 incorporation into SAGA, in

two highly divergent yeast species. Importantly, quantitative MS analyses show that Ada1 and Taf12 incorporation into SAGA does not require Spt20 (Supplementary Table 1). Nonetheless, it is formally possible that deleting or mutating the Spt20 HIT region affects Ada1 and/or Taf12 conformation and position within SAGA, such that their putative contacts with Tra1 are weakened.

Finally, analysis of SAGA conformations revealed continuous movements between Tra1 and the rest of SAGA, around the hinge[12].

**Fig. 7** The Spt20 HIT region is conserved and sufficient to incorporate Tra1 into SAGA. **a** Schematic illustration of the different Spt20 truncation mutants constructed to identify the Head Interacting with Tra1 (HIT) region of Spt20 from *S. cerevisiae*. Distinct colours depict Spt20 domains, defined as Homology Boxes (HB) and a Low Complexity Region (LCR). Each mutant is named after to the last residue present in the truncation mutant, which shortens the LCR to various extent, as illustrated. **b** Silver staining of *S. cerevisiae* SAGA complexes purified from WT and *spt20* truncation mutants, using Spt7 as the bait. Below is an anti-HA western blot of HA-Spt7-TAP in a fraction of the input used for TAP. Data are representative of three independent experiments. Hashtag indicates a doublet band likely resulting from an artefact of silver staining. In *S. cerevisiae*, the HIT region of Spt20 is narrowed down to residues 474–492 (purple-coloured box) and predicted to fold into a α-helix by PSIPRED[70] (left). **c** LC-MS/MS analyses of *S. cerevisiae* SAGA complexes purified from WT, *spt20–492*, *spt20–380*, *spt20–408*, and *spt20–474* truncation mutants, as in panel **b**. LFQ intensity ratios of Tra1 were normalised to the bait, Spt7. Ratios from two independent experiments are plotted individually with the mean (black bar). **d** Western blot analyses of V5-tagged Spt20 truncation mutants in SAGA complex eluates from panel **b**. **e** GST pull-down of *S. pombe* protein extracts from WT, *tra1-Sptra2*, and *tra1-Sctra1* strains, using GST fused to *S. pombe* (Sp) Spt20 HIT region (residues 282–324) (top right panel) or GST fused to *S. cerevisiae* (Sc) Spt20 HIT region (residues 468–537) (bottom right panel). GST alone serves as a negative control for background binding to glutathione beads. Anti-FLAG western blotting is used to detect WT and hybrid Tra1 proteins bound to GST or GST-HIT columns (right panel) and in a fraction (0.6%) of the input used for the pull-downs. Data that are representative of three independent experiments. Coomassie blue staining of purified GST fusion proteins are shown on the right. Source data are provided as a Source Data file

This observation may have important implications for the allosteric regulation of SAGA activities. Indeed, such structural flexibility suggests that the interaction between Tra1 and Spt20 is dynamic. Therefore, depending on the conformation of the entire complex, Tra1 might directly interact with subunits other than Spt20, including Ada1 and Taf12. Understanding the molecular basis and the functional relevance of this flexibility is an important goal of future research projects, but it will require innovative methodological approaches. Overall, our findings open new perspectives to understand the molecular mechanism by which Tra1 modulates SAGA enzymatic activities upon binding transcription activators.

Recent seminal work established that complexes are generally assembled by ordered pathways that appear evolutionarily conserved[1]. Biochemical analyses of SAGA in various mutants suggested that the last steps of SAGA assembly occur through an ordered pathway. Indeed, Spt20 is required for both Tra1 and DUB incorporation into SAGA, while Tra1 stabilises the DUB, but not Spt20. Conversely, the DUB module does not regulate Spt20 or Tra1 assembly. Finally, monitoring the fate of the DUB component Sgf11 upon Tra1 de novo synthesis supports a model in which Tra1 interacts with Spt20 and then promotes incorporation of the DUB module (Fig. 8f). However, Tra1 is presumably not directly recruiting the DUB module into SAGA. Recent structural analyses indicate that Tra1 does not stably contact any DUB component in most mature SAGA conformations[12]. Rather, Tra1 might stabilise DUB incorporation during the assembly process, either through transient direct interaction or indirectly, by inducing a conformational change within Spt20 that allows SAGA-DUB interactions. Combining our work with previous structural and biochemical analyses suggests that Spt20 directly contacts the DUB anchor subunit, Sgf73, although a higher resolution structure of SAGA is needed to validate this hypothesis.

To conclude, these results contribute to our understanding of the mechanisms and pathways by which multifunctional transcription complexes assemble, which is essential to characterise their structural organisation and regulatory roles. The current model for Tra1 function postulates that it transmits the transactivation signal from promoter-bound transcription factors to SAGA and NuA4 activities, which have critical roles in both basal and inducible RNA polymerase II transcription. Along this line, we noted that only a small number of genes require Tra1, Tra2 and Spt20 for their expression in *S. pombe* (Figs. 1a, b, 5e, f, 6g–i, and Supplementary Fig. 3). However, our RNA-seq analyses measured transcript levels at steady state. Recent work using nascent RNA-seq demonstrated that SAGA has a much more global role on RNA polymerase II transcription in *S. cerevisiae*[49]. Therefore, it will be important for future studies to use such methodological approaches, in order to determine the genome-wide effects of SAGA and NuA4 on transcription rates in

*S. pombe*. Our work opens exciting prospects for the characterisation of Tra1 exact roles during transcription initiation and of its specific contribution to SAGA and NuA4 regulatory activities.

## Methods

**Yeast manipulation and growth conditions**. Standard culture media and genetic manipulations were used. *S. cerevisiae* strains were grown in YPD at 30 °C to mid-log phase (~$1 \times 10^7$ cells per ml). *S. pombe* strains were grown in either rich (YES) or minimal (EMM) media at 32 °C to mid-log phase (~$0.5 \times 10^7$ cells per ml). Proliferation assays were performed by inoculating single colonies in liquid media and counting the number of cells at different time points. For longer time courses, cultures were diluted to keep cells in constant exponential growth. Cell viability was assessed using 10 μL of the colorimetric dye methylene blue (319112, Sigma), which was incubated with a 50 μL suspension of exponentially growing yeast cells resuspended in PBS 1×. The number of blue-coloured dead cells was counted under a light microscope. For auxin-inducible targeted protein degradation (AID), cells were grown at 25 °C and treated with either 0.5 mM indol-3-acetic acid (IAA, I2886, Sigma) or ethanol. For CreER-loxP-mediated recombination, cells were treated with either 1 μM β-estradiol (E2758, Sigma) or DMSO, for either 18 h (*tti2-CKO* strains) or 21 h (*tra2-CKO* and control *creER* strains), unless otherwise indicated.

**Strain construction**. All *S. pombe* and *S. cerevisiae* strains used are listed in Supplementary Table 4 and were constructed by standard procedures, using either chemical transformation or genetic crosses. Strains with gene deletions, truncations or C-terminally epitope-tagged proteins were constructed by PCR-based gene targeting of the respective open-reading frame (ORF) with kanMX6, natMX6 or hphMX6 cassettes, amplified from pFA6a backbone plasmids[50,51]. For insertion of loxP sites, the same resistance cassettes were amplified from the pUG6 or pUG75 plasmids (Euroscarf #P30114, and #P30671, respectively)[52]. Constructions of point mutations, internal deletions or domain swaps in *spt20* and *tra1* were performed using a *ura4* cassette in a two step in vivo site-directed mutagenesis procedure[53]. Alternatively, CRISPR-Cas9-mediated genome editing was used, as described[54], for example for marker-less N-terminal epitope tagging of *tra1*. DNA fragments used for homologous recombination were generated by PCR, Gibson assembly cloning (kit E2611L, New England Biolabs), or gene synthesis. Cloning strategies and primers were designed using the online fission yeast database, PomBase[55]. All relevant primer sequences are listed in Supplementary Table 5. Transformants were screened for correct integration by PCR and, when appropriate, verified by Sanger sequencing or Western blotting. For each transformation, 2–4 individual clones were purified and analysed.

Because the *tti2* gene is essential for viability in *S. pombe*[22], C-terminal epitope tagging was performed in diploids, to generate heterozygous alleles. Their sporulation demonstrated that all C-terminally tagged Tti2 strains grew similarly to wild-type controls in all conditions that were tested.

**Plasmid construction**. Auxin-inducible degron (AID) tagging was performed using a plasmid, DHB137, which we constructed by inserting three HA epitopes in fusion with the three copies of the mini-AID sequence from pMK151[56]. V5-PK tagging was performed using a plasmid, DHB123, which we constructed by inserting three V5 epitopes 5′ to the hphMX6 cassette into pFA6a-hphMX6 (Euroscarf #P30438)[51]. For GST pull-down assays, DNA fragments comprising either nucleotides +925 to +1054 from the *S. pombe spt20* ORF, encoding residues Asp282 to Ala324, or nucleotides +1402 to +1611 from the *S. cerevisiae SPT20* ORF, encoding residues Met468 to Ala537, were synthesised and amplified. Each product was then subcloned into pGEX-4T2 (GE Healthcare Life Sciences), 3′ and in frame to the GST coding sequence, using the Gibson assembly kit (E2611L, New England Biolabs), to generate the DHB179 and DHB193 plasmids, respectively.

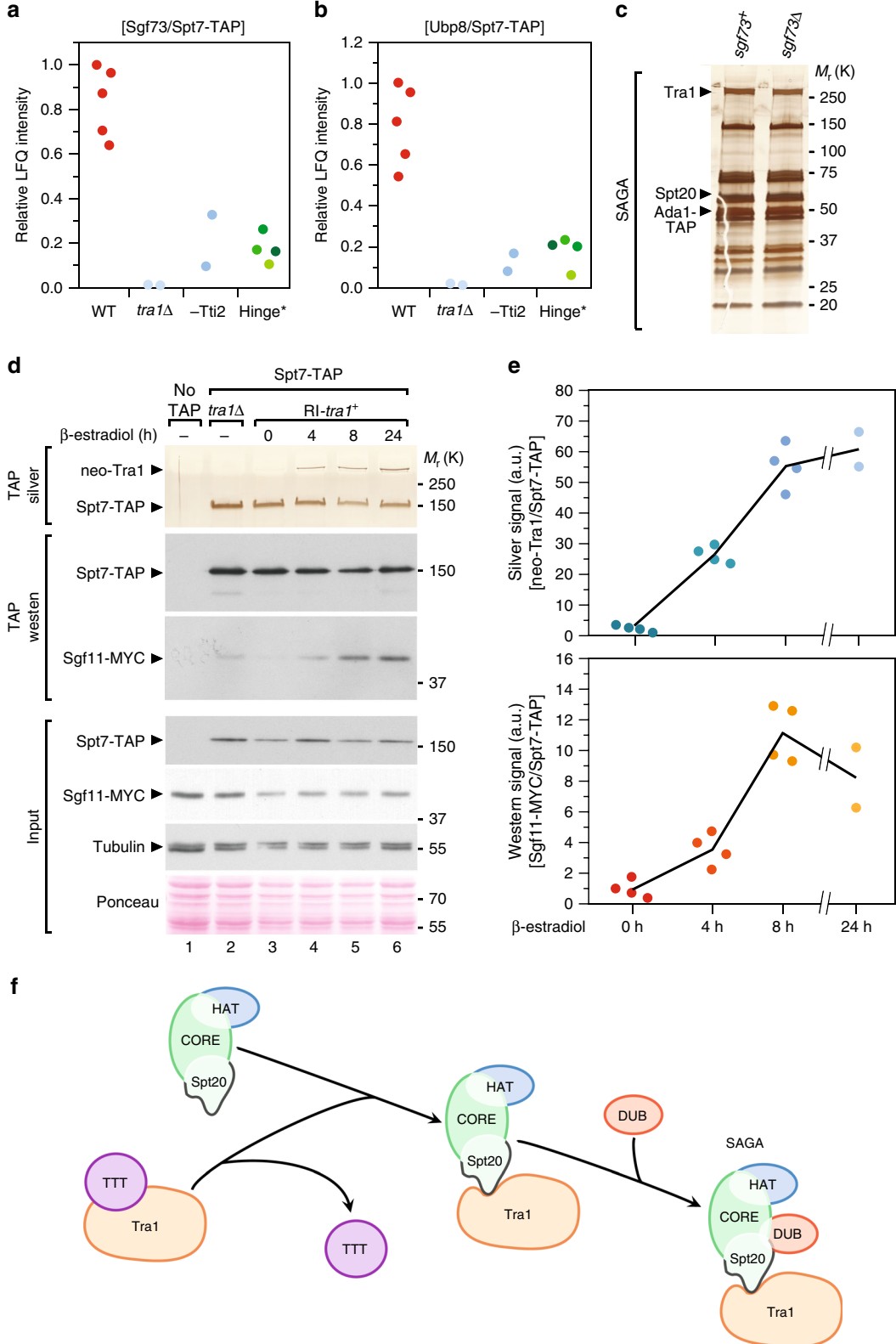

**RT-qPCR**. Reverse transcription and quantitative PCR analyses of cDNA were performed using RNA extracted from 50 mL of exponentially growing cells, as described[57], and according to the MIQE guidelines[58]. Briefly, the total RNA was purified using hot, acidic phenol, and contaminating DNA was removed by DNase I digestion, using the TURBO DNA-free™ kit (AM1907, Ambion). One microgram of RNA was then reverse-transcribed (RT) at 55 °C with random hexanucleotide primers, using the SuperScript III First-Strand System (18080051, Thermo Fisher

Scientific). Fluorescence-based quantitative PCR was performed with SYBR Green and used to calculate relative cDNA quantities, from the slope produced by standard curves for each primer pair, generated in each experiment. DNase-treated RNA samples were used as controls for the presence of genomic DNA contaminants. Standard curve slopes were comprised between −3.5 (90% efficiency) and −3.15 (110% efficiency), with an $r^2 > 0.9$. All primer sequences are listed in Supplementary Table 5.

**Fig. 8** Tra1 promotes DUB module assembly into SAGA. **a**, **b** LC-MS/MS analysis of SAGA complexes purified from mutants defective in Tra1-SAGA interaction, including *tra1Δ* (*n* = 2) and β-estradiol-treated *tti2-CKO* mutants (-Tti2, *n* = 2, grown as indicated in Fig. 2a), as well as four distinct mutants that disrupt the hinge region (Hinge). These include *tra1-Sptra2*, *spt20–290*, *spt20-HITΔ* and *spt20-FIEN* mutants, labelled from dark to light green, respectively. Relative LFQ intensity ratios of Sgf73 (**a**) and Ubp8 (**b**) to the bait, Spt7, from independent experiments or mutants are plotted individually. **c** Silver staining of SAGA complexes purified from WT (*sgf73*) or *sgf73Δ* mutants, using Ada1 as the bait. Data are representative of three independent experiments. **d** Silver staining and western blotting of SAGA complexes purified upon Tra1 synthesis (neo-Tra1) from a strain in which the DUB subunit Sgf11 is MYC-tagged. *spt7-TAP Rl-tra1 sgf11-MYC* cells were grown to exponential phase and harvested at different time points after β-estradiol addition, as indicated (hours). SAGA was purified from a *tra1Δ* strain as a control for the complete loss of Tra1 from SAGA and from a non-tagged strain (no TAP) as a control for background. Silver staining reveals Spt7 and Tra1, which migrate around 150 and 400 kDa, respectively. Anti-HA and anti-MYC western blotting of Spt7-TAP and Sgf11-MYC in a fraction of the input (Input) and in TAP eluates (TAP) is shown below. An anti-tubulin antibody and Ponceau red staining are used as loading controls. Data are representative of four independent experiments, quantified and averaged in (**e**). **e** Quantification of the ratio of Tra1 to Spt7 from silver stained gels (top) and of the ratio of Sgf11-MYC to Spt7-TAP from western blots (bottom). Signal intensities were quantified from four independent experiments (*n* = 4), except at 24 h (*n* = 2). Each data point was plotted individually. **f** Working model for the last steps of SAGA assembly. The core subunits (Spt7, Ada1 and TAFs), the HAT module (Gcn5, Ada2, Ada3 and Sgf29) and Spt20 form a pre-assembled complex. The Hsp90 cochaperone TTT promotes the maturation of nascent Tra1, which is then anchored to SAGA by the HIT domain of Spt20. Consequently, Tra1 stabilises the interaction of the DUB module (Sgf73, Ubp8, Sgf11 and Sus1) with SAGA to form a mature, fully active multifunctional transcriptional co-activator complex. Source data are provided as a Source Data file

**Protein extraction**. Protein extracts were prepared as described[57]. Briefly, 10–25 mL cultures of exponentially growing cells were homogenised by glass bead-beating in a FastPrep (MP Biomedicals). Proteins were extracted using either a standard lysis buffer (WEB: 40 mM HEPES-NaOH pH 7.4, 350 mM NaCl, 0.1% NP40 and 10% glycerol) or trichloroacetic acid (TCA) precipitation. WEB was supplemented with protease inhibitors, including complete EDTA-free cocktails tablets (04693132001, Roche), 1 mM PMSF (P7626, Sigma), 1 μg per ml bestatin (B8385, Sigma) and 1 μg per ml pepstatin A (P5318, Sigma). Protein concentrations were measured by the Bradford method. Ponceau red or Coomassie blue staining were used to normalise for total protein levels across samples.

**Western blotting and antibodies**. Western blotting was performed using the following antibodies: peroxidase-anti-peroxidase (PAP) (P1291, Sigma, 1:2000); anti-Calmodulin binding protein (CBP) (RCBP-45A-Z, ICLab, 1:500); anti-tubulin (B-5–1–2, Sigma, 1:5000); anti-FLAG (M2, F1804, Sigma, 1:1000); anti-MYC (9E10, Agro-Bio LC; 9E11, ab56, Abcam, 1:1000; or rabbit polyclonal ab9106, Abcam, 1:1000); anti-V5 (SV5-Pk1, AbD Serotec, 1:1000); anti-HA (16B12, Ozyme; rabbit polyclonal, ab9110, Abcam, 1:1000). Protein concentrations were measured by the Bradford method and used to load equal amounts of proteins across samples. Quantification of signal intensity was performed using staining, film exposure or digital acquisition that were within the linear range of detection, as verified by loading serial dilutions of one sample, and analysed using ImageJ.

**Chromatin immunoprecipitation**. ChIP experiments were performed as previously described[11]. Briefly, cell cultures were cross-linked in 1% formaldehyde for 30 min. Cells were then broken using a FastPrep (MP Biomedicals), and the chromatin fraction was sheared to 200–500 bp fragments using a Branson sonicator for nine cycles (10 s ON, 50 s OFF) at an amplitude of 20%. For immunoprecipitation (IP), 3–5 μg of anti-HA (16B12) or anti-Myc antibodies (9E11) were incubated overnight at 4 °C with the chromatin extracts and then coupled with 50 μl of protein-G-sepharose beads (GE17–0618–01, Sigma) during 4 h at 4 °C. ChIP DNA was quantified by fluorescence-based quantitative PCR using SYBR Green, as described for RT-qPCR analysis. Input (IN) samples were diluted 200-fold, while IP samples were diluted threefold. Relative occupancy levels were determined by dividing the IP by the IN value (IP/IN) for each amplicon. To determine the specificity of enrichment of the tagged protein, the corresponding untagged control samples were included in each ChIP experiment. All primer sequences are listed in Supplementary Table 5.

**Affinity purification**. Protein complexes were purified by the tandem affinity purification (TAP) method, as described previously[39,59], with minor modifications. One to four liters of exponentially growing cells were harvested, snap-frozen as individual droplets and grinded in liquid nitrogen using a Freezer/Mill® (Spex SamplePrep). Protein extraction was performed in either WEB buffer or CHAPS-containing lysis buffer (CLB) buffer (50 mM HEPES-NaOH pH 7.4, 300 mM NaCl, 5 mM CHAPS, 0.5 mM DTT), supplemented with protease and phosphatase inhibitors. Following purifications, 10% of 2 mM EGTA eluates were concentrated and separated on 4–20% gradient SDS-polyacrylamide Tris-glycine gels (Biorad). Total protein content was visualised by silver staining, using the SilverQuest kit (LC6070, Thermo Fisher Scientific). For quantitative mass spectrometry analyses, 40% of 2 mM EGTA eluates were precipitated with TCA and analysed by mass spectrometry (MS). A downscaled version of the TAP procedure was used for standard co-immunoprecipitation followed by western blot analysis, as described previously[57].

Recombinant GST and GST-HIT proteins were produced by IPTG induction of transformed BL21 Rosetta strains and purified on 100 μl of Glutathione Sepharose 4B beads (17075601, GE Healthcare Life Sciences), for 4–5 h at 4 °C. After washing, beads were further incubated overnight at 4 °C with 5–10 mg of *S. pombe* protein extracts prepared in WEB lysis buffer, before analysis by Coomassie blue staining and western blotting.

**Mass spectrometry and data analysis**. Dry TCA precipitates from TAP eluates were denatured, reduced and alkylated. Briefly, each dry TCA precipitate sample was dissolved in 89 μL of TEAB 100 mM, and 1 μL of DTT 1 M was added before incubation for 30 min at 60 °C. A volume of 10 μL of IAA 0.5 M was added (incubation for 30 min in the dark). Enzymatic digestion was performed by addition of 1 μg trypsin (Gold, Promega, Madison USA) in TEAB 100 mM and incubation overnight at 30 °C. After completing the digestion step, peptides were purified and concentrated using OMIX Tips C18 reverse-phase resin (Agilent Technologies Inc.) according to the manufacturer's specifications. Peptides were dehydrated in a vacuum centrifuge.

Samples were resuspended in 9 μL formic acid (0.1%, buffer A), and 2 μL were loaded onto a 15 cm reversed phase column (75 mm inner diameter, Acclaim Pepmap 100® C18, Thermo Fisher Scientific) and separated with an Ultimate 3000 RSLC system (Thermo Fisher Scientific) coupled to a Q Exactive Plus (Thermo Fisher Scientific) via a nano-electrospray source, using a 143-min gradient of 2–40% of buffer B (80% ACN, 0.1% formic acid) and a flow rate of 300 nl per min.

MS/MS analyses were performed in a data-dependent mode. Full scans (375–1500 m $z^{-1}$) were acquired in the Orbitrap mass analyser with a 70,000 resolution at 200 m $z^{-1}$. For the full scans, $3 \times 10^6$ ions were accumulated within a maximum injection time of 60 ms and detected in the Orbitrap analyser. The 12 most intense ions with charge states ≥ 2 were sequentially isolated to a target value of $1 \times 10^5$ with a maximum injection time of 45 ms and fragmented by HCD (Higher-energy collisional dissociation) in the collision cell (normalised collision energy of 28%) and detected in the Orbitrap analyser at 17,500 resolution.

Raw spectra were processed using the MaxQuant environment (v1.5.0.0 or v1.5.5.1)[60] and Andromeda for database search with label-free quantification (LFQ), match between runs and the iBAQ algorithm enabled[61]. The MS/MS spectra were matched against the UniProt reference proteome (Proteome ID UP000002485) of *S. pombe* (strain 972 / ATCC 24843) (Fission yeast) (release 2017_10, https://www.uniprot.org/), 250 frequently observed contaminants, as well as reversed sequences of all entries. Different release versions were used, depending on the date of analysis (Supplementary Table 6). Enzyme specificity was set to trypsin/P, and the search included cysteine carbamidomethylation as a fixed modification and oxidation of methionine, and acetylation (protein N-term) and/or phosphorylation of Ser, Thr, Tyr residue (STY) as variable modifications. Up to two missed cleavages were allowed for protease digestion. FDR was set at 0.01 for peptides and proteins and the minimal peptide length at 7.

The relative abundance of proteins identified in each affinity purification was calculated as described[62]. Briefly, label-free quantification (LFQ) intensity values were transformed to a base 2 logarithmic scale (Log2), to fit the data to a Gaussian distribution and enable the imputation of missing values. For Tti2 interactome analysis (Supplementary Data 1), normalised LFQ intensities were compared between replicates, using a 1% permutation-based false discovery rate (FDR) in a two-tailed Student's *t* test. The threshold for significance was set to 1 (fold change = 2), based on the ratio between TAP and no-TAP samples and the FDR. The relative abundance of subunits in each purification eluate was obtained by dividing the LFQ intensity of that interactor (prey) to the LFQ intensity of the TAP purified

protein (bait). For clarity purposes, this ratio was further normalised to that obtained in control strains in Fig. 5, Fig. 6, Fig. 7 and Supplementary Fig. 9.

**RNA-seq and data analysis**. Each growth condition or yeast strain was analysed in triplicate. RNA was extracted from 50 mL of exponentially growing cells RNA using TRIzol reagent (15596018, Thermo Fisher Scientific). DNA was removed by DNase I digestion, using the TURBO DNA-free™ kit (AM1907, Ambion), and RNA was cleaned using the RNeasy Mini kit (74104, Qiagen). The total RNA quality and concentration was determined using an Agilent Bioanalyzer. Transcripts were purified by polyA-tail selection. Stranded dual-indexed cDNA libraries were constructed using the Illumina TruSeq Stranded mRNA Library Prep kit. Library size distribution and concentration were determined using an Agilent Bioanalyzer. 48 libraries were sequenced in one lane of an Illumina HiSeq 4000, with $1 \times 50$ bp single reads, at Fasteris SA (Plan-les-Ouates, Switzerland). After demultiplexing according to their index barcode, the total number of reads ranged from 6 to 10 million per library.

Adapter sequences were trimmed from reads in the Fastq sequence files. Reads were aligned using HISAT2[63], with strand-specific information (–rna-strandness R) and otherwise default options. For all 48 samples, the overall alignment rate was over 95%, including over 90% of reads mapping uniquely to the *S. pombe* genome. Reads were then counted for gene and exon features using htseq-count[64] in union mode (–mode union), reverse stranded (–stranded Reverse), and a minimum alignment quality of 10 (–minaqual 10). For all samples, over 95% of reads were assigned to a feature (–type gene). Variance-mean dependence was estimated from count tables and tested for differential expression based on a negative binomial distribution, using DESeq2[65]. Pairwise comparison or one-way analysis of variance were run with a parametric fit and genotype as the source of variation (factor: mutant or control). Hierarchical clustering of genes which expression changes in control *creER*, Tti2-depleted, Tra2-depleted and Tra1 knockout cells was performed on the Log2 fold changes calculated using DESeq2. Genes were clustered based on Pearson correlation distance measurements and using the complete linkage method, in Cluster 3.0[66]. All other computational analyses were run either on R (version 3.6.0), using publicly available packages, or on the Galaxy web platform using the public server at usegalaxy.org[67].

**Statistical analysis**. All statistical tests were performed using either R (version 3.6.0) for RNA-seq data or GraphPad Prism for proteomic data. All other experiments were analysed using GraphPad Prism. *t* tests were used when comparing two means. One-way or two-way analyses of variance (ANOVA) were performed for comparing more than two means, across one (for example "genotype") or two distinct variables (for example "genotype" as a between-subject factor and "time" as a within-subject factor). One-way and two-way ANOVAs were followed by Tukey and Bonferroni post-hoc pairwise comparisons, respectively. An α level of 0.01 was used a priori for all statistical tests, except otherwise indicated. Comparisons that are statistically significant ($P \leq 0.01$) are marked with one asterisk.

**Reporting summary**. Further information on research design is available in the Nature Research Reporting Summary linked to this article.

## Data availability
The raw sequencing data reported in this publication have been deposited in NCBI Gene Expression Omnibus[68] and are accessible through GEO Series accession number GSE128448. The mass spectrometry proteomics data have been deposited to the ProteomeXchange Consortium via the PRIDE[69] partner repository with the data set identifier PXD013256. Supplementary Table 6 cross-references all MaxQuant results from each figure or table with the raw data files. All other relevant data supporting the key findings of this study are available within the article and its Supplementary Information files or from the corresponding author upon reasonable request. The source data underlying all figures and supplementary figures, including uncropped versions of all gels and blots and the raw data of all reported graphs, are provided as a Source Data file. A reporting summary is available as a supplementary information file.

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

## Acknowledgements

We acknowledge Kerstin Wagner and Elsa Cesari for invaluable technical assistance and all members of the Helmlinger laboratory for helpful suggestions and discussions. We thank Patrick Schultz and Adam Ben Shem for kindly sharing results before their publication. We thank Fred Winston for critical reading of the manuscript and sharing *S. cerevisiae* strains. We are grateful to Robin Allshire, Charles Hoffman, Hisao Masukata, and Paul Russell for sharing *S. pombe* strains. A.E.V. is a recipient of a post-doctoral fellowship from the Fondation pour la Recherche Médicale (SPF20130526854). D.T. is a recipient of a graduate fellowship from the French Ministry for Research and Higher Education (MESR), supported by the Labex EpiGenMed, an « Investissements d'avenir » program (ANR-10-LABX-12–01). This work was supported by funds from the CNRS (ATIP-Avenir), the FP7 Marie Curie Actions (FP7-PEOPLE-2012-CIG/COACTI-VATOR), and the Agence Nationale de la Recherche (ANR-15-CE12–0009–01 and ANR-15-CE11–0022–03) to D.H.

## Author contributions

A.E.V., D.T. and C.F. designed, performed and analysed all experiments, except RNA-seq analyses (D.H.) and quantitative LC-MS/MS experiments (M.S.); D.H. designed the study, acquired funding and supervised the project; D.H. wrote the paper with input from all authors.

## Competing interests

The authors declare no competing interests.
