## [Peer Review File · Nature Communications]

Reviewers' comments:

Reviewer #1 (Remarks to the Author):

The conserved protein Tra1 is essential for viability in *S. cerevisiae* and mammalian cells, due to its function in multiple complexes. However, in *S. pombe*, there are two Tra proteins, Tra1 and Tra2, and Tra1 is not essential, thus facilitating its study *in vivo*. In this manuscript, the authors present experiments that address the assembly of Tra1 and Tra2 into the *S. pombe* SAGA and NuA4 complexes, respectively. Their work provides strong evidence that assembly into each complex requires chaperones, consistent with previous studies in mammalian cells. Furthermore, their studies provide new information regarding the assembly of Tra1 into SAGA, demonstrating that it requires a small region of Spt20, another SAGA component. Finally, they provide evidence that Tra1 is required for normal levels of the DUB module of SAGA in the mature complex. Overall, the results in this manuscript are quite interesting and should be of general interest to those who study transcription and coactivators. Comments to be addressed are listed below.

1. The manuscript requires editing throughout to correct the English. Here are three early examples (I'm only showing the corrections): page 2, line 10 – assembles; page 2, line 17 – in this process; page 3, line 26 – indicate that the primary role of Tra1...These types of changes are needed throughout the manuscript. The authors should find someone to carefully edit the English.
2. Also, with respect to the writing –the word “remarkably” is used too often. I would take a look at each use and consider other words.
3. Figure 1B – Instead of, or in addition to the Venn diagram, it would be informative to show a scatterplot comparing the different datasets. This would provide more quantitative information than the Venn diagram with respect to possible similarities between the different conditions. The authors refer to the overlap as remarkable – so please provide some statistical analysis of the overlap.
4. For the experiments using estradiol-treated cells to excise the *tra2+* or *tii2+* loci, for how long were the cells treated in each experiment? And, importantly, what was the viability of the cells at the time at which RNA was extracted? Please provide the viability data. Finally, please provide quantitation for the depletion of Tti2.

5. page 6, line 7 – Do the authors mean Tra1 affects SAGA and NuA4 recruitment, or do they mean Tra1 for SAGA and Tra2 for NuA4? In any case, references should be cited for this statement about the role of Tra1/2 in recruitment.

6. page 7, Figure 2, Figure S5 – Why are the authors using Tel2 instead of Tti2 for these experiments? As for the Cre/Lox experiments, what was the viability of the cells after 16 hours of IAA treatment? Please provide quantitation for the degree of Tel2-AID depletion.

7. Figure 2D, 2F – There is no statistical analysis done for the experiments in these panels. The authors should provide information on the number of times these were done and their reproducibility.

8. Fig. 2B and Fig. 3 – The authors should clarify how they are quantifying the amount of NuA4 purified in the +Tti2/-Tti2 and +Tra2/-Tra2 conditions. Are they basing this on loading 10% of eluates? How can they rule out that they are merely getting less protein from sick or dying cells? It would be very helpful to have a control, such as purification of SAGA after Tra2 depletion. Otherwise, this does not seem like a strong conclusion.

9. Fig. 4C – Are we supposed to conclude anything about the higher level of SAGA in lane 2?

10. page 10, bottom – Did the authors test *ada1* and *taf12* mutants also?

11. Figure 5 – What happens to Tra1 protein levels in these mutants?

12. Figure 6A – Can the bands in these TAP preps be labeled to identify which bands correspond to which SAGA subunits? Also, what is the doublet band just below 37 kd in the spt20-380 prep?

13. While this paper presents compelling information regarding the specificity of Tra1 versus Tra2 in *S. pombe*, and also shows convincingly what region of Spt20 directly interacts with Tra1 and is sufficient for its recruitment into SAGA, I'm not sure we have gained any insight into what determines the distribution of one Tra1 pool between SAGA and NuA4 in *S. cerevisiae* or in mammalian cells. If the authors think they have provided some new information on this topic, it would be good to mention it in the Discussion.

Reviewer #2 (Remarks to the Author):

Here, the authors investigate mechanisms governing the assembly of the multi-subunit transcription factor complexes SAGA and NuA4, focusing on the role of the related and largest subunits of the complexes, Tra1/2. The authors use of the *S pombe* system, with Tra1 specific for SAGA and Tra2 specific for NuA4, allows an elegant way to determine the unique contributions of Tra to each complex and to decipher mechanisms that encode specificity for either SAGA or NuA4. Understanding the assembly pathway and principals governing assembly and function of large multi subunit complexes is an emerging and important field.

The authors find that the PIKK chaperone complex TTT is required for incorporation of Tra1/2 into SAGA and NuA4. This provides a possible mechanism for conservation of Tra1 structure even though the function of Tra1 and the other PIKKs have diverged considerably. They also find that a known TTT co chaperone, Hsp90, is also important. The effects of this latter factor are fairly modest, but this may be due to limitations in what Hsp90 alleles can be tolerated. Also related to assembly mechanisms, they find that Tra1 is important for recruitment of the SAGA DUB module, while Tra2 seems important but not absolutely essential for NuA4 assembly. Thus, the two highly related factors play distinct roles in assembly of the two large complexes. In a set of elegant studies, they also localize the region of Tra1 that confers specificity for assembly into SAGA and also find a functionally conserved region of SAGA subunit Spt20 that is responsible for this interaction. In sum, this manuscript reveals an important set of new information on assembly, function, and specificity of SAGA, NuA4 and its Tra1/2 subunits.

The authors should consider the following in a revised manuscript

1. In Fig 1a, the gene expression results in different strains have been clustered, but it is difficult to interpret from the data shown. Where are the boundaries of the clusters? It may be more informative for understanding the behavior of these clusters to show results as a box plot of RNA data from the different strains in addition to or instead of the heatmap.
2. Fig 2D is confusing. Why is the level of Tra1 lower than WT in lane 2 (with an intact TTT complex). It may help to compare the Tra1/Spt7 ratio in lane 2 vs WT but this value is not given.

3. The number of genes affected by mutations in Tra1, Tra2 and Tti are very modest. In *S. cerevisiae*, it was shown that SAGA mutations had a stabilizing effect on mRNA such that nascent RNA analysis was required to determine the genome-wide role of SAGA. I don't think it's reasonable for the authors to repeat all of their experiments using this newer approach, as the relevant points are made with steady state RNA. However, going forward it may be very informative for the authors to measure nascent transcription in SAGA and NuA4 mutants rather than steady state RNA – giving a clearer picture of the in vivo roles for both complexes.

Reviewer #3 (Remarks to the Author):

This manuscript is very interesting as it attempts to disentangle the fundamental process of the molecular assembly of Tra1 within the SAGA and Nu4A complex. Here, I am engaged specifically to evaluate the mass spectrometry data but I would like to also include some questions and comments for other scientific content. The authors have applied the MaxQuant interface for computational proteomics but they have not specified if the ensuing statistics for IP-MS data was performed using Perseus, a software that is often used after MaxQuant and was developed by the same group which developed MaxQuant. Or alternatively, the authors used Graphpad Prism for both Proteomics and RNA-Seq data.

(i) On page 6, line 24:

“We next tested if Tti2 prevents Tra1 and Tra2 disassembly from their complex or, rather, promotes their de novo incorporation.”

Also, On page 7, line 13:

“These results demonstrate that TTT contributes to the de novo incorporation of Tra1 into the SAGA complex.”

>> While the results from Figure 2D shows that TTT contributes to the de novo incorporation of Tra1 into the SAGA complex, I do not see, in addition to this, any evidence to support or refute the statement that “...Tti2 prevents Tra1 and Tra2 disassembly from their complex”. Can the authors explain?

(ii) On page 8, line 7:

“We noted that the absence of Tti2 affected SAGA and NuA4 differently. Upon Tti2 depletion, the decrease of Tra1 does not affect SAGA overall migration profile, similar to what we observed in a tra1 \square mutant (Figure 2A). In contrast, the effect of Tti2 on Tra2 incorporation within NuA4 is less pronounced, but seems to cause a global decrease in the amount of purified NuA4 (Figure 2B). Alternatively, the bait used for this purification, the Mst1 HAT 11 subunit, might dissociate from the rest of the complex upon tti2+ deletion and loss of Tra2.”

>> From Figure 2B, it does not seem to overall decrease in Nu4A. This is because the bait, i.e. Mst-TAP, also appears to be weaker in silver-stained gels and the Western blot.

(iii) On page 9, line 2:

“... how Tra1 interacts specifically with SAGA, taking advantage of the viability of tra1 mutants in S. pombe ...”

>> Are these so-named tra1 mutants referring specifically to the tra1 \square mutants? If yes, then there is a typo mistake here, so need to replace tra1 mutants with tra1 \square mutants instead. If they are referring to the S. pombe harboring tra1 mutations in general, then it is correct.

Now, coming back to the proteomics data.

(i) On page 24, line 12: “Dry TCA precipitates from TAP eluates were denatured, reduced and alkylated. Briefly, each 12 sample was dissolved in 89 μ L of TEAB 100 mM...”

>> The authors did not indicate any denaturing agents being in the buffer composition, but only TEAB solution.

(ii) On page 25, line 3:

>> 3 x 10⁶ ions should be 3 x 10⁶ or 3E6.

(iii) I noticed that during database search, variable modifications such as phosphorylations (STY) are used. Is it used to improve protein coverage, or to look for meaningful phosphorylation sites? If not, the inclusion of pSTY during search would simply enlarge the search space unnecessarily, leading to higher number of errorous identification.

(iv) The authors applied an arbitrary cut-off at 2-fold change. Setting 2-fold change as cut-off is not ideal as sometimes due to low abundance (low signals) and low reproducibility, this 2-fold change cut-off can be met easily. Actually, Perseus has a function for permutation-based FDR estimation to determine the outliers. This function considers both fold-changes and p-values (not fold-change alone). In this function, the LFQ values are first transformed to logarithm (log2), and the resulting Gaussian distribution of the data was used for imputation of missing values by normal distribution (width = 0.3, shift = 2.5). Statistical outliers were then determined using a two-tailed t test followed by multiple testing corrections with a permutation-based FDR method.

(v) While I am impressed with the wealth and systematic data generated by the authors using the IP-MS technique, I noticed that in the ProteomExchange repository, there is no cross-reference text file that allows readers to associate MaxQuant results to the RAW data files, as well as to which experiments as stated in the manuscript. It would be better to have such a file so that this allows us to reanalyze the raw data based on the results and discussion in the manuscript.

(vi) There are three versions of *S. pombe* databases used.

RefProteome_SPOMBE-cano_2016_11.fasta

RefProteome_SPOMBE-cano_2017_01.fasta

RefProteome_SPOMBE-cano_2017_10.fasta

In the manuscript, only 2017_10 is mentioned. Were the other two FASTA used for any analysis in this manuscript?

Tek Yew, Low

Responses to reviewer comments.

Reviewer #1 (Remarks to the Author):

The conserved protein Tra1 is essential for viability in S. cerevisiae and mammalian cells, due to its function in multiple complexes. However, in S. pombe, there are two Tra proteins, Tra1 and Tra2, and Tra1 is not essential, thus facilitating its study in vivo. In this manuscript, the authors present experiments that address the assembly of Tra1 and Tra2 into the S. pombe SAGA and NuA4 complexes, respectively. Their work provides strong evidence that assembly into each complex requires chaperones, consistent with previous studies in mammalian cells. Furthermore, their studies provide new information regarding the assembly of Tra1 into SAGA, demonstrating that it requires a small region of Spt20, another SAGA component. Finally, they provide evidence that Tra1 is required for normal levels of the DUB module of SAGA in the mature complex. Overall, the results in this manuscript are quite interesting and should be of general interest to those who study transcription and coactivators. Comments to be addressed are listed below.

We thank the reviewer for the encouraging evaluation of our manuscript.

- 1. The manuscript requires editing throughout to correct the English. Here are three early examples (I'm only showing the corrections): page 2, line 10 – assembles; page 2, line 17 – in this process; page 3, line 26 – indicate that the primary role of Tra1... These types of changes are needed throughout the manuscript. The authors should find someone to carefully edit the English.*

Response: We apologize for these errors and thank the reviewer for pointing them to us. A native English speaker has carefully read the manuscript, which we have edited accordingly. In particular, we simplified the last sentence of the abstract, which we agree was unnecessarily complicated. We kept “to this process” at the end of this sentence though (page 2, line 17), because we meant “contribution” as in “a contribution to something, to help make it efficient or successful”.

- 2. Also, with respect to the writing –the word “remarkably” is used too often. I would take a look at each use and consider other words.*

Response: We apologize for repeating the word “remarkably”, which we agree was not always used appropriately. Of the five occurrences, we changed it to “surprisingly” (page 4, line 17), to “notably” (page 13 line 7), or removed it (page 15, line 10).

- 3. Figure 1B – Instead of, or in addition to the Venn diagram, it would be informative to show a scatterplot comparing the different datasets. This would provide more quantitative information than the Venn diagram with respect to possible similarities between the different conditions. The authors refer to the overlap as remarkable – so please provide some statistical analysis of the overlap.*

Response: The rationale for performing RNA-seq analyses of *ttr2*, *tra2*, and *tra1* mutant strains was to compare their overall transcriptome profiles. We agree that scatter plots and correlation analysis provide more quantitative information and have remodeled Figure 1 and

Supplementary Figure 3 accordingly. The description of these RNA-seq results now starts with density scatter plots and Pearson correlation coefficient calculations (r statistics), shown in a novel Figure 1a, comparing the transcriptome profiles of *tii2*, *tra2*, and *tra1* mutants. Pairwise comparisons revealed a positive correlation in gene expression changes observed between *tii2* and *tra2* mutants (Figure 1a, left graph) or between *tii2* and *tra1* mutants (Figure 1a, middle). In contrast, as expected from their specific interaction with either SAGA or NuA4, gene expression changes in *tra2* and *tra1* mutants show no correlation (Figure 1a, right). We then show the number of differentially expressed genes (DEGs) using area-proportional Venn diagrams (Figure 1b). Statistical analysis of the overlap was calculated using a hypergeometric test and the resulting P values are shown in the legend to Figure 1b.

We complement these genome-wide studies with quantitative RT-PCR analyses of individual genes, which expression requires either both Tti2 and Tra1 (Figure 1c, formerly Supplementary Figure 3e) or both Tti2 and Tra2 (Figure 1d, formerly Supplementary Figure 3f).

Finally, we moved our hierarchical clustering analyses from Figure 1a to Supplementary Figure 3b,c, to show a more comprehensive and informative analysis of these clusters. Notably, as suggested by Reviewer #2 (see below), we now provide violin plots revealing the behavior of genes in each cluster and in all four conditions (novel Supplementary Figure 3c). We thank both Reviewers 1 and 2 for their constructive suggestions about how to better describe and analyze the transcriptome profiles of the *tii2*, *tra2*, and *tra1* mutants.

4. For the experiments using estradiol-treated cells to excise the tra2+ or tii2+ loci, for how long were the cells treated in each experiment? And, importantly, what was the viability of the cells at the time at which RNA was extracted? Please provide the viability data. Finally, please provide quantitation for the depletion of Tti2.

Response: Thank you for pointing this. The information was indeed missing from parts of the manuscript and we have carefully corrected this. Briefly, throughout our study, *tii2*-CKO strains were treated for 18 hours with DMSO or β -estradiol, while *tra2*-CKO strains were treated for 21 hours. Control *cre-ER* strains were treated for the longest time, 21 hours, with DMSO or β -estradiol. This information now also appears in the first section of the Materials and Methods section and is stated in the legends to all relevant figures (Figure 1-3) and supplementary figures (Supplementary Figure 1-3). In time-course experiments, the duration of β -estradiol addition is indicated directly on the figure (Figure 2c and 7d).

It is indeed important to verify that all biochemical (TAP purifications) and functional (RNA-seq, RT-qPCR) assays are performed using cultures of mostly viable yeast cells. We first identified which time point to use, based on the proliferation rate of *tii2*-CKO and *tra2*-CKO strains, which starts to decrease 18 and 21 hours after β -estradiol addition, respectively (Supplementary Figure 1d and 2d). However, as pointed out, this assay does not measure viability. We thus now provide cell viability estimates, which we measured using a colorimetric dye, methylene blue, and counting blue-coloured dead cells under a light microscope. This analysis revealed no decrease of cell viability in *tii2*-CKO strains treated with β -estradiol for 18 hours and a small, 14% decrease of cell viability in *tra2*-CKO strains treated with β -estradiol for 21 hours (shown in novel Supplementary Figures 1e and 2e). To conclude, all experiments presented in this manuscript were done using cultures containing mostly viable yeast cells. We have edited the

first paragraph of the Results section to include these data and clarified the experimental procedure used to analyse *tii2-CKO* and *tra2-CKO* strains. We thank the reviewer for bringing this important control to our attention.

Finally, Supplementary Figure 1 has been revised and now includes a quantification of Tti2 depletion from Western blot experiments. Quantifications were performed in ImageJ, using exposures acquired within the linear range of the chemiluminescence signal on an Amersham 600 imager.

5. page 6, line 7 – Do the authors mean *Tra1* affects SAGA and NuA4 recruitment, or do they mean *Tra1* for SAGA and *Tra2* for NuA4? In any case, references should be cited for this statement about the role of *Tra1/2* in recruitment.

Response: This sentence refers to the well-recognized role of *S. cerevisiae* Tra1 in contacting diverse transcription activators and recruiting SAGA or NuA4 to specific promoters. Therefore, in *S. pombe*, the model is that Tra1 recruits SAGA whereas Tra2 recruits NuA4 to chromatin. We have clarified this sentence and added references that support this statement.

6. page 7, Figure 2, Figure S5 – Why are the authors using *Tel2* instead of *Tti2* for these experiments? As for the *Cre/Lox* experiments, what was the viability of the cells after 16 hours of IAA treatment? Please provide quantitation for the degree of *Tel2-AID* depletion.

Response: We used an inducible depletion allele of *Tel2* for several reasons. First, we have shown that Tti2 forms a stable complex, called TTT, with *Tel2* and Tti1 (Supplementary Table 1). Conversely, affinity purification and quantitative mass spectrometry analysis of *Tel2* and Tti1 confirmed stoichiometric interactions between all three proteins (see figure below).

These results are part of another manuscript that we will submit for publication soon and are consistent with previous work performed in *S. pombe*, *S. cerevisiae*, and mammalian cells. These studies are referenced at the beginning of the Results section.

Second, in human cells, work from several laboratories indicates that the knock-down of Tti2 phenocopies that of Tel2 or Tti1 (see, for example, Izumi N *et al.*, *Cancer Sci.*, 2012). One possible explanation for these observations is that TTI2 is critical for the stability of both TEL2 and TTI1 at steady state (Hurov KE *et al.*, *Genes & Dev.*, 2010).

Third, for the experiments shown in Figure 2d, we had to switch to an inducible depletion system that does not rely on CreER activity, because we used it to follow the fate of Tra1 upon *de novo* synthesis. We therefore constructed strains in which each TTT subunit is fused to an auxin-inducible degron. Unfortunately, we observed that fusing Tti2 to an AID caused its destabilization even in absence of auxin. In contrast, Tel2-AID and Tti1-AID strains showed efficient auxin-induced depletion. The corresponding paragraph of the Results section has been amended to briefly explain the reason for switching from Tti2 to Tel2.

Furthermore, we now provide cell viability estimates measured using methylene blue staining, as previously explained (Reviewer #1, comment #4). We observed no decrease of cell viability in *tel2-AID* strains treated with auxin for 16 hours (shown in a novel Supplementary Figure 5c), indicating that the biochemical analyses presented in Figure 2c,d were done using viable cells. Again, we thank the reviewer for bringing this important control to our attention.

Finally, Supplementary Figure 5d has been revised and now includes a quantification of Tel2 depletion from Western blot experiments. Quantifications were performed as explained in our response to comment #4 from Reviewer #1.

7. Figure 2D, 2F – There is no statistical analysis done for the experiments in these panels. The authors should provide information on the number of times these were done and their reproducibility.

Response: The experiments in these panels were done in duplicates as stated in the legends and thus no statistical tests were performed. To show the reproducibility of these experiments explicitly, we have modified Figure 2 such that values obtained in each independent experiment are now plotted in dedicated graphs (Figure 2f). The left and right graphs show quantitative MS analyses of the purifications from which one silver-stained gel is shown in Figure 2d and 2e, respectively. Consequently, SAGA purifications from *hsp90-26* strains were moved from former Figure 2e to a novel Supplementary Figure 6.

8. Fig. 2B and Fig. 3 – The authors should clarify how they are quantifying the amount of NuA4 purified in the +Tti2/-Tti2 and +Tra2/-Tra2 conditions. Are they basing this on loading 10% of eluates? How can they rule out that they are merely getting less protein from sick or dying cells? It would be very helpful to have a control, such as purification of SAGA after Tra2 depletion. Otherwise, this does not seem like a strong conclusion.

Response: Our conclusion that Tra2 has a scaffolding role for the NuA4 complex should be indeed strengthened, as pointed by this reviewer and by Reviewer #3 (comment #2).

a) Quantification of purified complexes using mass spectrometry:

The procedure used to quantify protein-protein interactions from mass spectrometry data is detailed in the last paragraph of the “*Mass spectrometry and data analysis*” section of Materials and Methods and is identical to a pipeline previously established by the Vermeulen group (Smits AH *et al.*, *Nucleic Acids Res.*, 2013) and available in MaxQuant (v.1.5.5.1).

Briefly, for NuA4, we determined the abundance of each subunit co-purifying with the bait upon depletion of Tti2 (Figure 2b) and Tra2 (Figure 3), in three steps. First, we calculated the intensity-based absolute quantification (iBAQ) values in each purification eluate, to determine the relative abundance of all proteins within one sample and identify which protein is enriched. This analysis reproducibly identified all 13 NuA4 subunits, as shown for example in Supplementary Figure 7b. Second, we computed label-free quantification (LFQ) intensities to obtain an accurate measurement of the relative abundance of NuA4 subunits across different samples, while minimizing technical biases. Third, the LFQ intensities of each subunit was divided by the LFQ intensity of the bait (Mst1-TAP in Figure 2b or Vid21-TAP in Figure 3b) and these normalized values were then directly plotted using GraphPad Prism.

In parallel to LC-MS/MS analyses, 10% of the same purification eluate were loaded on a gradient SDS-PAGE and stained with silver nitrate. These gels are shown to allow a qualitative evaluation of each purification using a method orthogonal to quantitative mass spectrometry.

b) Total protein content upon Tti2 or Tra2 conditional depletion:

As suggested, we verified that all biochemical purifications were performed in optimal experimental conditions. First, SAGA and NuA4 were purified from *tii2-CKO* and *tra2-CKO* strains treated with either DMSO or β -estradiol for 18 and 21 hours, respectively. As shown in the novel Supplementary Figure 1e and 2e, these cultures contain mostly viable yeast cells (see our answer to comment #4 for details). Second, we did not observe a global decrease of total protein content from these cultures, as shown by Coomassie blue (novel Supplementary Figure 1f and 2f) and Ponceau red staining (amended Figure 3a) of total protein extracts. Last, Figure 3a now also includes Western blotting analysis of the expression of each bait (Epl1-TAP and Vid21-TAP) in the extracts used for tandem affinity purification.

c) Control SAGA purification upon Tra2 conditional depletion:

Finally, as suggested, we tested whether we can recover an intact chromatin-bound complex from cells that progressively arrest proliferating, by purifying SAGA in these experimental conditions, *ie.* 21 hours after adding β -estradiol to *tra2-CKO* cells. Silver staining analyses of purified Spt7-TAP eluates revealed no differences in SAGA subunit composition upon conditional loss of Tra2 (novel Supplementary Figure 8, which shows one representative result out of two independent experiments). This control experiment indicates that, at this time point, Tra2 depletion does not artefactually affects the extraction, purification, or stability of a chromatin complex that does not contain Tra2.

To conclude for this point, we thank the reviewer for these excellent suggestions, which were indeed important to strengthen our conclusion about the scaffolding role of Tra2 within the NuA4 complex. All novel data are incorporated within existing figures or in a novel Supplementary Figure 8, and described in the revised version of the manuscript, mostly within the “*Tra1 and Tra2 have distinct architectural roles between SAGA and NuA4*” section of the Results.

9. Fig. 4C – Are we supposed to conclude anything about of the higher level of SAGA in lane 2?

Response: We did not reproducibly observe higher levels of SAGA in Spt7-TAP purification eluates from *tra1-Sptr2* mutant strains (lane 2, Figure 4c), out of 4 independent experiments. We decided to show this replicate because, despite higher levels of SAGA in this lane, no Tra1 is detectable, further supporting our conclusion that the Tra1-SpTra2 hybrid mutant cannot interact with SAGA.

10. page 10, bottom – Did the authors test *ada1* and *taf12* mutants also?

Response: We did not test *ada1* or *taf12* mutants because there were biochemical and functional evidence that these subunits have a more global role in SAGA architecture and integrity than Spt20. Work in *S. cerevisiae* demonstrated that Ada1 is a core architectural component of SAGA (Wu PY and Winston F, *Mol. Cell Biol.*, 2002) and heterodimerizes with Taf12 through histone-fold domains (Gangloff YG *et al.*, *Mol. Cell Biol.*, 2000). Taf12 itself is also an integral subunit of the general transcription factor TFIID, with which SAGA presumably shares an octamer of histone folds at its core. Accordingly, we previously showed that deleting *ada1* causes severe growth defects and *taf12* deletion is lethal in *S. pombe* (Helmlinger D *et al.*, *The EMBO Journal*, 2011).

It is possible though that a specific domain or a discrete motif from Ada1 and/or Taf12 directly contacts Tra1, possibly in distinct conformations of SAGA. Higher resolution structures of different SAGA conformers are needed to address this point specifically, which remains a challenging task. Regardless, our structure-function characterization of Spt20 from both *S. pombe* and *S. cerevisiae* indicate that, if Ada1 and/or Taf12 directly interact with Tra1, their contribution to Tra1 incorporation into SAGA is minor in these experimental conditions, as compared to the necessary and sufficient role of the Spt20 HIT region. We have amended the Discussion to discuss the possible contribution of Ada1 and Taf12.

11. Figure 5 – What happens to Tra1 protein levels in these mutants?

Response: We analyzed Tra1 levels in all *spt20* mutants from Figure 5. In absence of a specific antibody, we crossed all *spt20* mutant strains with a strain in which we fused a FLAG epitope to the N-terminus of endogenous Tra1. Western blot analyses revealed that Tra1 protein levels remain similar to those observed in control WT cells (novel panel b in Supplementary Figure 10).

Together with our observation that the Tra1-SpTra2 hybrid mutant is also expressed normally (Figure 4c), we conclude that unassembled Tra1 is stable and correctly folded, contrary to core SAGA subunits, which are highly unstable in absence of their partners (Wu PY and Winston F, *Mol. Cell Biol.*, 2002). These results are consistent with the peripheral position of Tra1 in the latest structure of SAGA and with the observation that Tra1 structure is similar whether in isolation or bound to SAGA (Cheung ACM and Diaz-Santin LM, *Transcription*, 2018). We describe these novel data at the end of the second paragraph within “*The SAGA subunit Spt20 anchors Tra1 into the SAGA complex*” section of the Results.

12. *Figure 6A – Can the bands in these TAP preps be labeled to identify which bands correspond to which SAGA subunits? Also, what is the doublet band just below 37 kd in the spt20-380 prep?*

Response for labeling bands on SAGA purifications: Although this information would indeed be particularly useful to have, we were unfortunately unable to confidently label most bands on the silver stained gel of purified *S. cerevisiae* SAGA complexes. The predicted molecular weight of several SAGA subunits falls within a narrow range (50-60 kDa) and all LC-MS/MS analyses were performed in liquid solution, using TAP eluates. In our hands, this procedure gives better sequence coverage and thus more accurate quantifications than in-gel trypsin digestion, which, conversely, would have allowed us to identify the subunits present in each band.

Response for the doublet band: We did not reproducibly observe this doublet band below 37kDa in SAGA purification eluates from *spt20-380* mutants, out of 3 independent experiments. We amended Figure 6 by adding a silver staining analysis of *S. cerevisiae* SAGA purified from another clone of *spt20-380* mutants, together with a new truncation mutant, *spt20-408*. We have labeled this doublet and amended the legend to Figure 6 to describe what is likely an artifact from silver staining.

13. *While this paper presents compelling information regarding the specificity of Tra1 versus Tra2 in S. pombe, and also shows convincingly what region of Spt20 directly interacts with Tra1 and is sufficient for its recruitment into SAGA, I'm not sure we have gained any insight into what determines the distribution of one Tra1 pool between SAGA and NuA4 in S. cerevisiae or in mammalian cells. If the authors think they have provided some new information on this topic, it would be good to mention it in the Discussion.*

Response: Indeed, our study does not directly provide a mechanism for how *S. cerevisiae* Tra1 or human TRRAP incorporates into either SAGA or NuA4/TIP60. This issue is the focus of ongoing work in our laboratory.

Taking advantage of Tra1 duplication and divergence in *S. pombe*, we previously reported that Tra1 and Tra2 are biochemically non-redundant. Indeed, we never detected Tra2 peptides in SAGA complexes purified from *tra1Δ* mutants (Helmlinger D. *et al.*, *The EMBO Journal*, 2011). In the present study, we initially hypothesized that the Hsp90 cochaperone TTT controls whether Tra1 is incorporated into either complex. However, no Tra2 peptides were detected in SAGA purified from Tti2-depleted cells (Figure 2a). To further test this possibility, we repeated these purifications from strains in which both Tti2 and Tra1 are absent. Again, no Tra2 peptides were detected in LC-MS/MS analyses of SAGA purified from β -estradiol-treated *spt7-TAP tra1Δ creER tti2-CKO* cells. Altogether these results suggest that Tra1 assembles into either SAGA or NuA4 independently of its maturation by TTT.

[Redacted]

What determines the distribution of Tra1 between SAGA and NuA4 then? One possibility is that SAGA and NuA4 subunits compete for binding to Tra1 CSI region. This mechanism would be similar to that described for mTOR assembly into either the TORC1 or TORC2 complexes. Recent structural analyses established that the TORC1-specific subunit Raptor and the TORC2-

specific subunit Rictor compete for binding to the same region of mTOR (Yang H *et al*, *Cell*, 2016; Karuppasamy M *et al.*, *Nat. Commun.*, 2017). Supporting this possibility, recent structural analyses of yeast NuA4 using electron microscopy and cross-linking coupled to MS, revealed that Tra1 FAT domain makes extensive contacts with several NuA4 subunits (Wang X *et al.*, *Nat. Commun.*, 2018; Setiaputra D *et al.*, *Mol. Cell. Biol.*, 2018). These interactions would sterically prevent Spt20 from recognizing the 3 α -helices forming the Tra1 CSI region. Higher resolution structures and further *in vitro* biochemical studies are required to directly test this possibility and explain why Tra1 binding to SAGA and NuA4 is mutually exclusive. We amended the Discussion with a novel paragraph discussing this hypothesis, at the end of the “*Distinct architectural roles of Tra1 between the SAGA and NuA4 complexes*” section.

Reviewer #2 (Remarks to the Author):

*Here, the authors investigate mechanisms governing the assembly of the multi-subunit transcription factor complexes SAGA and NuA4, focusing on the role of the related and largest subunits of the complexes, Tra1/2. The authors use of the *S pombe* system, with Tra1 specific for SAGA and Tra2 specific for NuA4, allows an elegant way to determine the unique contributions of Tra to each complex and to decipher mechanisms that encode specificity for either SAGA or NuA4. Understanding the assembly pathway and principals governing assembly and function of large multi subunit complexes is an emerging and important field.*

The authors find that the PIKK chaperone complex TTT is required for incorporation of Tra1/2 into SAGA and NuA4. This provides a possible mechanism for conservation of Tra1 structure even though the function of Tra1 and the other PIKKs have diverged considerably. They also find that a known TTT co chaperone, Hsp90, is also important. The effects of this latter factor are fairly modest, but this may be due to limitations in what Hsp90 alleles can be tolerated. Also related to assembly mechanisms, they find that Tra1 is important for recruitment of the SAGA DUB module, while Tra2 seems important but not absolutely essential for NuA4 assembly. Thus, the two highly related factors play distinct roles in assembly of the two large complexes. In a set of elegant studies, they also localize the region of Tra1 that confers specificity for assembly into SAGA and also find a functionally conserved region of SAGA subunit Spt20 that is responsible for this interaction. In sum, this manuscript reveals an important set of new information on assembly, function, and specificity of SAGA, NuA4 and its Tra1/2 subunits.

We thank the reviewer for the encouraging evaluation of our manuscript.

1. In Fig 1a, the gene expression results in different strains have been clustered, but it is difficult to interpret from the data shown. Where are the boundaries of the clusters? It may be more informative for understanding the behavior of these clusters to show results as a box plot of RNA data from the different strains in addition to or instead of the heatmap.

Response: As suggested, we have revisited the heatmap analysis of these RNA-seq data. We now identify 7 distinct clusters with clear boundaries (revised panel now in Supplementary Figure 3b). In addition, we show the behavior of all transcripts within each cluster in the different mutant strains using violin box plots (novel Supplementary Figure 3c). For clarity and to incorporate the suggestions from Reviewer #1 (comment #3), we have moved this analysis to Supplementary Figure 3, which was substantially modified. We thank the reviewer for this excellent suggestion, which has undoubtedly improved our presentation of the RNA-seq data.

2. Fig 2D is confusing. Why is the level of Tra1 lower than WT in lane 2 (with an intact TTT complex). It may help to compare the Tra1/Spt7 ratio in lane 2 vs WT but this value is not given.

Response: In lane 2 of Figure 2d, the level of Tra1 is low despite a functional TTT complex because we analyzed the amount of newly synthesized Tra1 in purified SAGA complexes, rather than Tra1 steady-state levels. In the experiments shown in Figure 2d and 2e, SAGA was purified from *RI-tra1* strains treated with β -estradiol for only 6 hours. As shown in Figure 2c, at this time point, newly synthesized Tra1 levels still gradually increase within SAGA.

We have not quantified the amount of newly synthesized Tra1 in SAGA at this time point, as compared to its steady-state levels (the 'WT' lane in Figure 2c and Figure 2d, shown as a positive control). However, out of 4 independent time-course experiments, we reproducibly detected less Tra1 six hours after β -estradiol addition than at steady-state. We thus chose this time point to examine the contribution of TTT (Figure 2d) and Hsp90 (Figure 2e) to the *de novo* assembly of Tra1 into SAGA. We apologize for the confusion and have clarified Figure 2 and its legend, notably by labeling newly synthesized Tra1 as 'neo-Tra1' in Figure 2c, 2d, and 2e.

3. *The number of genes affected by mutations in Tra1, Tra2 and Tti are very modest. In S. cerevisiae, it was shown that SAGA mutations had a stabilizing effect on mRNA such that nascent RNA analysis was required to determine the genome-wide role of SAGA. I don't think it's reasonable for the authors to repeat all of their experiments using this newer approach, as the relevant points are made with steady state RNA. However, going forward it may be very informative for the authors to measure nascent transcription in SAGA and NuA4 mutants rather than steady state RNA – giving a clearer picture of the in vivo roles for both complexes.*

Response: This is absolutely correct and a very relevant point to discuss. We have amended the Discussion to acknowledge this point and discuss the importance of using nascent approaches in future studies, in order to measure the rate of RNA synthesis in SAGA and NuA4 mutants, rather than steady-state transcript levels. This paragraph was added at the end of the Discussion.

Reviewer #3 (Remarks to the Author):

1. *This manuscript is very interesting as it attempts to disentangle the fundamental process of the molecular assembly of Tra1 within the SAGA and Nu4A complex. Here, I am engaged specifically to evaluate the mass spectrometry data but I would like to also include some questions and comments for other scientific content. The authors have applied the MaxQuant interface for computational proteomics but they have not specified if the ensuing statistics for IP-MS data was performed using Perseus, a software that is often used after MaxQuant and was developed by the same group which developed MaxQuant. Or alternatively, the authors used Graphpad Prism for both Proteomics and RNA-Seq data.*

Response: We thank the reviewer for the encouraging comment. All statistical analyses of AP-MS data were performed using GraphPad Prism. We have clarified the corresponding section “Statistical analysis” of Materials and Methods (page 29).

2. *(i) On page 6, line 24: “We next tested if Tti2 prevents Tra1 and Tra2 disassembly from their complex or, rather, promotes their de novo incorporation.” Also, On page 7, line 13: “These results demonstrate that TTT contributes to the de novo incorporation of Tra1 into the SAGA complex.” While the results from Figure 2D shows that TTT contributes to the de novo incorporation of Tra1 into the SAGA complex, I do not see, in addition to this, any evidence to support or refute the statement that “...Tti2 prevents Tra1 and Tra2 disassembly from their complex”. Can the authors explain?*

Response: Indeed, we did not directly test if the observed effect of TTT on Tra1 and Tra2 incorporation into SAGA and NuA4, respectively, results from a role in disassembly. However, many studies in different organisms have shown that TTT is an Hsp90 cochaperone dedicated to the stabilization of active PIKK kinases. Therefore, TTT would have to prevent Tra1 and Tra2 dissociation from either SAGA or NuA4 to explain the results shown in Figure 2.

3. *(ii) On page 8, line 7: “We noted that the absence of Tti2 affected SAGA and NuA4 differently. Upon Tti2 depletion, the decrease of Tra1 does not affect SAGA overall migration profile, similar to what we observed in a tra1D mutant (Figure 2A). In contrast, the effect of Tti2 on Tra2 incorporation within NuA4 is less pronounced, but seems to cause a global decrease in the amount of purified NuA4 (Figure 2B). Alternatively, the bait used for this purification, the Mst1 HAT 11 subunit, might dissociate from the rest of the complex upon tti2+ deletion and loss of Tra2.” From Figure 2B, it does not seem to overall decrease in Nu4A. This is because the bait, i.e. Mst-TAP, also appears to be weaker in silver-stained gels and the Western blot.*

Response: We indeed observed a decrease in the amount of the bait, Mst1-TAP, upon Tti2 depletion, in both total extracts (Western blot in Figure 2b) and NuA4 purification eluates (silver staining in Figure 2b). We have amended the corresponding section of the Results to acknowledge this caveat (page 9).

Please note that a reduction in subunit expression can be difficult to interpret because stability and protein-protein interaction are often tightly coupled, particularly for subunits of macromolecular complexes, such as NuA4. In addition, besides this issue, we realized that

Mst1 is not the ideal bait to assess NuA4 complex stability because it probably occupies a peripheral position within the complex and can exist outside of NuA4 as part of the Piccolo HAT module. Thus, to rigorously analyze the contribution of Tra2 to NuA4 subunit composition (Figure 3), we used Epl1 and Vid21 as baits, two subunits showing extensive contacts with other NuA4 components and with important roles in NuA4 architecture in *S. cerevisiae*.

4. (iii) On page 9, line 2: "... how Tra1 interacts specifically with SAGA, taking advantage of the viability of tra1 mutants in *S. pombe* ..." Are these so-named tra1 mutants referring specifically to the tra1D mutants? If yes, then there is a typo mistake here, so need to replace tra1 mutants with tra1D mutants instead. If they are referring to the *S. pombe* harboring tra1 mutations in general, then it is correct.

Response: We were indeed referring to the viability of *S. pombe* strains harboring tra1 mutations in general. Thank you for pointing this.

Now, coming back to the proteomics data.

5. (i) On page 24, line 12: "Dry TCA precipitates from TAP eluates were denatured, reduced and alkylated. Briefly, each 12 sample was dissolved in 89 μ L of TEAB 100 mM..." The authors did not indicate any denaturing agents being in the buffer composition, but only TEAB solution.

Response: We apologize for the confusion. DTT 1 M was used as a denaturing agent. This sentence was edited accordingly (page 26).

6. (ii) On page 25, line 3: 3×10^6 ions should be 3×10^6 or 3E6.

Response: Thank you for pointing this typo to us, which we have corrected. We identified and corrected a similar typo a few lines below.

7. (iii) I noticed that during database search, variable modifications such as phosphorylations (STY) are used. Is it used to improve protein coverage, or to look for meaningful phosphorylation sites? If not, the inclusion of pSTY during search would simply enlarge the search space unnecessarily, leading to higher number of erroneous identification.

Response: Indeed, we included a few post-translational modifications during database search, including STY phosphorylation, to increase the percentage coverage of each subunit. To minimize the number of erroneous identifications, which is indeed a caveat of this procedure, all our results were filtered at 1% false discovery rate (FDR) using MaxQuant. Last, except in the experiments shown in Supplementary Table 1 and Supplementary Figure 7, the goal of our LC-MS/MS analyses was to quantify the amount of SAGA subunits, mostly Tra1, in purified complexes rather than identifying new interacting partners.

8. (iv) The authors applied an arbitrary cut-off at 2-fold change. Setting 2-fold change as cut-off is not ideal as sometimes due to low abundance (low signals) and low reproducibility, this 2-fold change cut-off can be met easily. Actually, Perseus has a function for permutation-based FDR estimation to determine the outliers. This function considers both fold-changes and p-values (not fold-change alone). In this function, the LFQ values are first transformed to

logarithm (log2), and the resulting Gaussian distribution of the data was used for imputation of missing values by normal distribution (width = 0.3, shift = 2.5). Statistical outliers were then determined using a two-tailed t test followed by multiple testing corrections with a permutation-based FDR method.

Response: We indeed did not use this function from Perseus but have analyzed our data using a similar workflow, which is detailed in the last paragraph of the “*Mass spectrometry and data analysis*” section of Materials and Methods. We agree that a fold-change cut-off of 2 is arbitrary and have therefore revised Supplementary Table 1 to include all proteins identified in Tti2 AP-MS experiments. For each identified protein, numbers indicate the Log2 of LFQ intensity ratios [Tti2-TAP/no-TAP] and the *q* value calculated using a 1% permutation-based false discovery rate (FDR) in a two-tailed Student’s *t*-test.

9. *(v) While I am impressed with the wealth and systematic data generated by the authors using the IP-MS technique, I noticed that in the ProteomExchange repository, there is no cross-reference text file that allows readers to associate MaxQuant results to the RAW data files, as well as to which experiments as stated in the manuscript. It would be better to have such a file so that this allows us to reanalyze the raw data based on the results and discussion in the manuscript.*

Response: Thank you this excellent suggestion. We have generated a cross-reference file that will allow readers to link MaxQuant results to the RAW data files. Our revised manuscript now includes this file as supplementary information (Supplementary Table 4), which is referenced in the ‘*Data availability*’ section of Materials ad Methods.

10. *(vi) There are three versions of S. pombe databases used.*

RefProteome_SPOMBE-cano_2016_11.fasta

RefProteome_SPOMBE-cano_2017_01.fasta

RefProteome_SPOMBE-cano_2017_10.fasta

In the manuscript, only 2017_10 is mentioned. Were the other two FASTA used for any analysis in this manuscript?

Response: Indeed, several versions of *S. pombe* databases were used, depending on when each analysis was done. The cross-reference file (Supplementary Table 4) specifies which FASTA file was used for each analysis / figure. We modified the 4th paragraph of the “*Mass spectrometry and data analysis*” section of Materials and Methods to include this information.

REVIEWERS' COMMENTS:

Reviewer #1 (Remarks to the Author):

The authors have done an excellent job of addressing the comments. This manuscript makes a strong contribution that will be of great interest to the transcription community.

Reviewer #2 (Remarks to the Author):

The authors have done an excellent job of addressing my comments. The revised manuscript is an important contribution and I recommend publication.

Reviewer #3 (Remarks to the Author):

The authors have satisfactorily addressed most of my concerns and I support the publication of this manuscript. For comment #5, DTT is actually a reducing agent that breaks the disulphide bonds thus helps some degree in protein denaturation. I was expecting non-ionic agents such as urea being used, but since only DTT was used by the authors in this work. It is okay with me.